# GATA6 regulates WNT and BMP programs to pattern precardiac mesoderm during the earliest stages of human cardiogenesis

Joseph A Bisson[1], Miriam Gordillo[1], Ritu Kumar[1†], Neranjan de Silva[1], Ellen Yang[1], Kelly M Banks[1], Zhong-Dong Shi[2], Kihyun Lee[2‡], Dapeng Yang[2], Wendy K Chung[3], Danwei Huangfu[2], Todd Evans[1,4,5]*

[1]Department of Surgery, Weill Cornell Medicine, New York, United States; [2]Developmental Biology Program, Sloan Kettering Institute, New York, United States; [3]Childrens Hospital, Harvard Medical School, Boston, United States; [4]Hartman Institute for Therapeutic Organ Regeneration, Weill Cornell Medicine, New York, United States; [5]Center for Genomic Health, Weill Cornell Medicine, New York, United States

**\*For correspondence:**
tre2003@med.cornell.edu

**Present address:** †Gladstone Institutes, San Francisco, United States; ‡College of Pharmacy, Ewha Womans University, Seoul, South Korea

## eLife Assessment

This **important** study investigates the function of a critical regulator of human early cardiac development. The **convincing** examination of GATA6 function is thorough and well-executed. The study will be of interest to scientists working on how the human heart acquires its identity.

**Abstract** Haploinsufficiency for *GATA6* is associated with congenital heart disease (CHD) with variable comorbidity of pancreatic or diaphragm defects, although the etiology of disease is not well understood. Here, we used cardiac directed differentiation from human embryonic stem cells (hESCs) as a platform to study GATA6 function during early cardiogenesis. GATA6 loss-of-function hESCs had a profound impairment in cardiac progenitor cell (CPC) specification and cardiomyocyte (CM) generation due to early defects during the mesendoderm and lateral mesoderm patterning stages. Profiling by RNA-seq and CUT&RUN identified genes of the WNT and BMP programs regulated by GATA6 during early mesoderm patterning. Furthermore, interactome analysis detected GATA6 binding with developmental transcription factors and chromatin remodelers, suggesting cooperative regulation of cardiac lineage gene accessibility. We show that modulating WNT and BMP inputs during the first 48 hr of cardiac differentiation is sufficient to partially rescue CPC and CM defects in *GATA6* heterozygous and homozygous mutant hESCs. This study provides evidence of the regulatory functions for GATA6 directing human precardiac mesoderm patterning during the earliest stages of cardiogenesis to further our understanding of haploinsufficiency causing CHD and the co-occurrence of cardiac and other organ defects caused by human *GATA6* mutations.

## Introduction

Congenital heart disease (CHD) is the most common human birth defect, affecting nearly 1% of live births (*Hoffman and Kaplan, 2002*). CHD is generally understood to derive from dysfunction in the growth, morphogenesis, and/or differentiation of cardiac-fated progenitor cell derivatives; however,

**eLife digest** Congenital heart disease results in babies being born with structural defects to their hearts. It is one of the most common kind of human birth defects, yet its genetic causes remain poorly understood. Previous studies have identified a gene known as *GATA6*, which is sometimes altered in patients with the condition. As for most human genes, cells typically carry two *GATA6* copies, each inherited from one parent. In many congenital heart disease patients, only one of the two copies of the gene presents harmful mutations. However, it has so far remained difficult to investigate in the laboratory how this genetic profile results in heart defects.

To bypass these limitations, Bisson et al. used human stem cells derived from the early embryo (known as hESCs) that can become any tissue in the body, including the heart. The team genetically designed hESC lines carrying either one (heterozygous) or two (homozygous) mutant copies of *GATA6*. Experiments showed that homozygous cells failed to generate any cardiac cells, while those stemming from heterozygous cells were partially impaired. Further molecular analyses established that *GATA6* acts early in development by regulating WNT and BMP, two signaling pathways that contribute to hESCs becoming heart cells.

These findings indicate that embryos in which both copies of GATA6 are defective cannot generate heart cells, and therefore are not viable. They also suggest that modulating WNT and BMP pathways early during development may partially rescue heart defects in mutant embryos. Overall, the work by Bisson et al. offers a promising avenue for future research into congenital heart disease by providing researchers with hESC lines in which GATA6 is mutated.

the genetic and molecular networks governing these processes remain under investigation. Treatment options are limited to surgical interventions that can save the life of the infant but do not correct the underlying genetics. The causative genetic variants may contribute to complications and morbidity in adulthood (*Woudstra et al., 2017*). Research aimed to expand our understanding of the molecular genetics regulating normal human cardiogenesis is thus needed to advance therapeutic strategies in the detection and treatment of CHD.

Studies of cardiac development in model organisms have led to the discovery of key molecular drivers of heart organogenesis, including the GATA family of zinc-finger transcription factors. GATA factors, named by their ability to bind to the consensus sequence (*A/T*)GATA(*A/G*) (*Patient and McGhee, 2002*), represent six family members with GATA4/5/6 expressed in the developing heart of early-stage embryos (*Peterkin et al., 2007*; *Molkentin, 2000*). GATA4/5/6 transcription factors regulate multiple stages of heart development including the specification, proliferation, and differentiation of cardiac progenitor cells (CPCs) (*Peterkin et al., 2005*). Notably, *GATA6* knockout (KO) mice exhibit early embryonic arrest during primitive streak formation due to defects in visceral endoderm, a phenotype that has complicated the study of the early developmental function of GATA6 in vivo (*Morrisey et al., 1998*). Mice with conditional deletion of *GATA6* from early CPCs develop heart abnormalities including atrioventricular canal defects, ventricular septal defects, and partial loss of trabeculae (*Tian et al., 2010*; *van Berlo et al., 2010*). While these *GATA6* conditional homozygous KO mice die at birth due to heart malformations, few cardiac phenotypes were reported in the *GATA6* heterozygous mutants, except for recent reports of bicuspid aortic valve and heart rhythm abnormalities (*Gharibeh et al., 2018*; *Gharibeh et al., 2021*).

In humans, heterozygous mutations in *GATA6* are associated with various forms of CHD including outflow tract (OFT) defects, septal defects, and Tetralogy of Fallot (*Wang et al., 2012*; *Zhang et al., 2018*). GATA6 is required for the development of multiple mesoderm and endoderm-derived organs, and some CHD patients with heterozygous *GATA6* mutations have comorbidities including pancreatic agenesis, neonatal diabetes, or congenital diaphragmatic hernia (*Gong et al., 2013*; *Yu et al., 2014*). The phenotypic diversity of birth defects in patients containing *GATA6* mutations illustrates the complexity of human haploinsufficiency; there is likely a combination of variants in modifying or interacting genes that converge to influence the disease phenotype (*Fahed et al., 2013*; *Seidman and Seidman, 2002*). Previous studies of the function of *GATA6* in model organisms have not sufficiently modeled the variety of cardiac (and noncardiac) defects observed in humans and most describe phenotypes only after homozygous inactivation of the gene. This raises an important question; why do

CHDs in humans associate with *GATA6* haploinsufficiency? Species-specific functions for GATA6 may differ between mouse and humans, for example, raising the question of how observations made in model organisms relate to human cardiac development. There is a clear need to investigate the function of GATA6 in the context of a human system to better understand the basis for haploinsufficiency.

Current in vitro cardiac-directed differentiation protocols for human pluripotent stem cells (hPSCs) provide powerful tools for studying the transcriptional regulatory networks relevant to CHD (*Mummery et al., 2012*). A previous study using human induced pluripotent stem cells (hiPSCs) with CRISPR-Cas9-induced *GATA6* mutations showed that GATA6 is required at the stages of CPC and cardiomyocyte (CM) differentiation (*Sharma et al., 2020*). GATA6 is also required for definitive endoderm (DE) differentiation toward the pancreatic lineage as reported in several studies using hPSCs (*Shi et al., 2017*; *Heslop et al., 2021*; *Chia et al., 2019*; *Tiyaboonchai et al., 2017*). However, the function of GATA6 during early mesoderm patterning of human cardiac differentiation has not been explored, and how in vitro cardiac and DE lineage phenotypes caused by GATA6 loss-of-function relate to one another is not understood.

To determine the function of GATA6 during the earliest stages of cardiac differentiation, we performed in vitro CM-directed differentiation using *GATA6* loss-of-function human embryonic stem cell (hESC) lines and explored regulatory functions for GATA6 during precardiac mesoderm patterning that is ultimately required for CM generation. Transcriptome analysis revealed that *GATA6* mutations impaired expression of genes important for lateral and cardiac mesoderm patterning, and by integrating GATA6 CUT&RUN sequencing with RNA-seq we identified GATA6-regulated targets including those of the WNT and BMP networks, key morphogenetic signaling pathways required to induce anterior primitive streak formation (*Ivanovitch et al., 2021*; *Funa et al., 2015*). Furthermore, we found that GATA6 interacts with important developmental transcription factors and chromatin remodelers during early mesoderm patterning stages that may promote DNA accessibility required for cardiac lineage commitment. Taken together, these findings uncover multiple functions for GATA6 in regulating very early precardiac mesoderm patterning and the lineage-specific gene networks required for human CM generation.

## Results

### GATA6 loss-of-function impairs cardiac mesoderm lineage differentiation

A monolayer cytokine-based in vitro cardiac differentiation protocol for hPSCs was adapted from previously established methods (*Dubois et al., 2011*; *Kattman et al., 2011*; *Yang et al., 2022*; *Figure 1A*). *GATA6* heterozygous mutant (*GATA6$^{+/-}$*), homozygous mutant (*GATA6$^{-/-}$*), and isogenic wildtype (WT) control (*GATA6$^{+/+}$*) H1-hESC lines were previously generated (*Shi et al., 2017*; *González et al., 2014*; *Figure 1—figure supplement 1A and B*). These lines were differentiated in parallel to determine the impact of *GATA6* loss-of-function during human cardiogenesis. Western blotting at days 2 or 5 of cardiac differentiation confirmed the loss of GATA6 protein in *GATA6$^{-/-}$* cells and intermediate levels of protein in *GATA6$^{+/-}$* cells relative to WT controls (*Figure 1—figure supplement 1C*). The percentage of cardiac troponin T-positive (%cTnT$^{+}$) CMs in individual differentiation assays remained stable starting around day 13 and thus CM differentiation efficiency was analyzed during a window of days 13–18 of cardiac differentiation. *GATA6$^{-/-}$* cells failed to generate beating CMs and yielded close to 0% cTnT$^{+}$ CMs when examined by flow cytometry (*Figure 1B and C*). In contrast, *GATA6$^{+/-}$* hESCs did generate beating CMs (~25% cTnT$^{+}$) but at a significantly reduced efficiency compared to WT cells (~55% cTnT$^{+}$, *Figure 1B and C*). *GATA6$^{+/-}$* CMs obtained were comparable morphologically to WT CMs with no obvious sarcomere defects (*Figure 1—figure supplement 1D*). No significant differences in %cTnT$^{+}$ were observed between clones of the same genotype (*Figure 1C*); therefore, data from lines with the same genotypes were combined in all subsequent comparisons.

We also generated an iPSC line derived from a CHD patient presenting with an atrial septal defect and congenital diaphragmatic hernia; this patient was found to have a heterozygous frameshift mutation in *GATA6* (c.1071delG) (*Yu et al., 2014*). The mutant allele was corrected to WT sequence (*GATA6$^{corr/+}$*) using CRISPR-Cas9 gene editing (*Figure 1—figure supplement 1E and F*). The iPSC colonies expressed normal levels of the pluripotency markers NANOG and SOX2 demonstrating efficient reprogramming (*Figure 1—figure supplement 1G*). We performed cardiac differentiation using

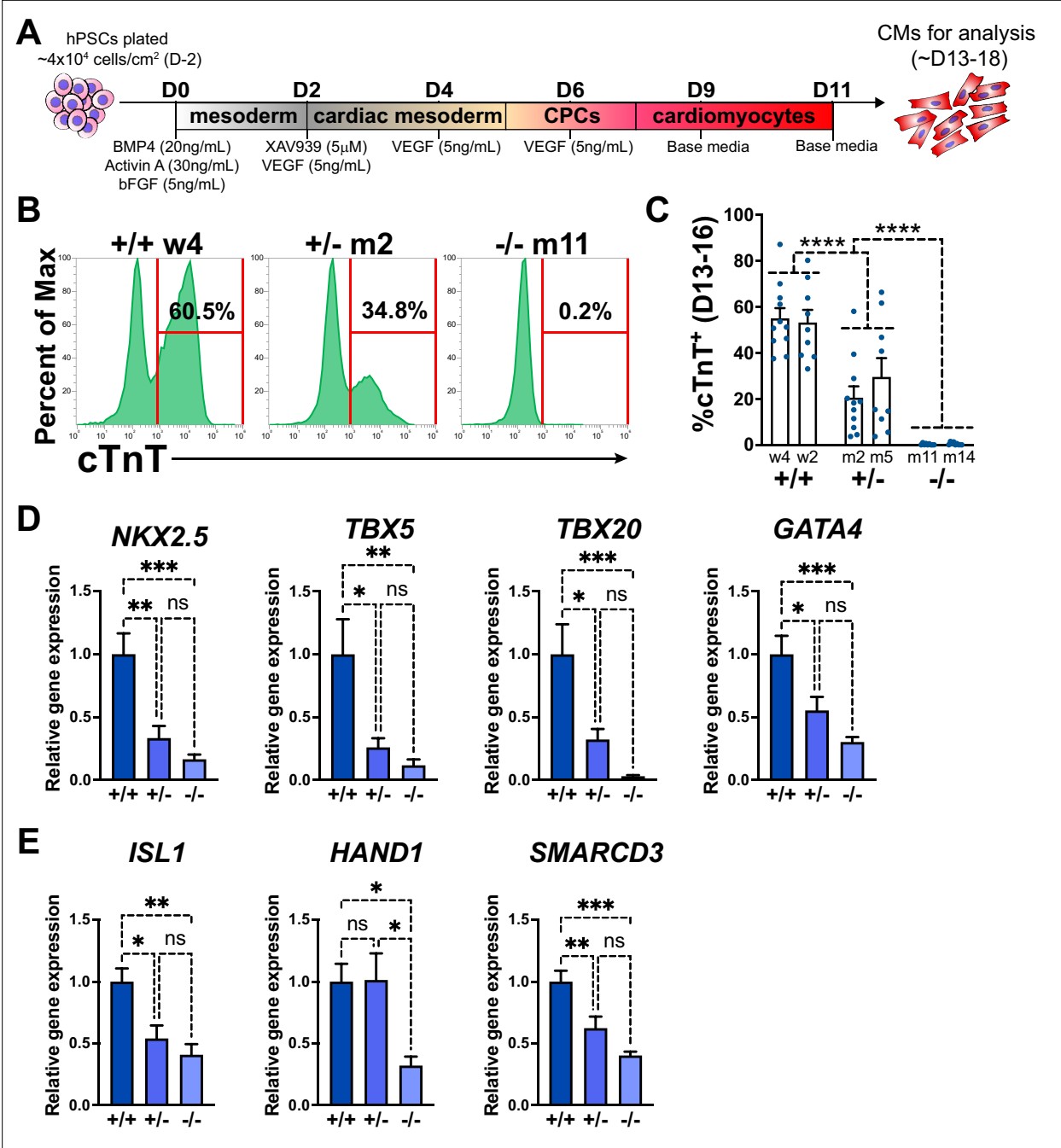

**Figure 1.** GATA6 loss-of-function inhibits cardiomyocyte (CM) and cardiac progenitor cell (CPC) development. (**A**) Schematic for in vitro CM-directed differentiation cytokine-based protocol (used throughout the study except for *Figure 1—figure supplement 2B–E*). Color gradients represent relative developmental stages (white: pluripotent; gray: mesoderm; yellow: cardiac mesoderm; pink: CPCs; and red: CMs). (**B**) Representative flow cytometry plots for cTnT+ CMs at day 14 of cardiac differentiation for *GATA6+/+*, *GATA6+/-*, and *GATA6-/-* hESCs. (**C**) %cTnT+ CMs quantified by flow cytometry between days 13 and 16 (dots indicate independent biological replicates). Significance indicated as ****p<0.0001 by two-way ANOVA with Tukey's multiple-comparison test by genotype. There was no significant difference between clones of the same genotype when two-way ANOVA and Tukey's multiple-comparison test were performed for all six sample groups. (**D**) Day 6 RT-qPCR for the CPC markers indicated normalized to *GATA6+/+* (n = 6). (**E**) Day 6 RT-qPCR for heart field markers normalized to *GATA6+/+* (n = 6). Data represents the mean ± SEM, significance indicated as *p<0.05, **p<0.01, ***p<0.001, ns indicates not significant as determined by one-way ANOVA and Tukey's multiple-comparison test. The labels w4, w2 (wildtype), m2, m5 (heterozygous), and m11, m14 (homozygous) refer to the isogenic wildtype and mutant hESC clones (see *Figure 1—figure supplement 1B*).

The online version of this article includes the following source data and figure supplement(s) for figure 1:

**Source data 1.** cTnT flow cytometry in panel C.

*Figure 1 continued on next page*

*Figure 1 continued*

**Source data 2.** RT-qPCR in panels D and E.

**Figure supplement 1.** CRISPR gene editing and characterization of mutant GATA6 human embryonic stem cell (hESC) and induced pluripotent stem cell (iPSC) lines.

**Figure supplement 1—source data 1.** Western blots in panels C and K.

**Figure supplement 1—source data 2.** cTnT flow cytometry in panel H.

**Figure supplement 1—source data 3.** Original unmarked western blot images in panel K.

**Figure supplement 2.** Cardiac progenitor cell (CPC) marker gene transcriptional analysis of GATA6 WT or mutant human embryonic stem cells (hESCs).

**Figure supplement 2—source data 1.** RT-qPCR in panels A and F.

the cytokine-based protocol for $GATA6^{corr/+}$ and $GATA6^{1071delG/+}$ iPSCs to determine how the phenotype compared to the $GATA6^{+/+}$ and $GATA6^{+/-}$ H1-hESCs. To date, this is the first patient-derived iPSC line with a heterozygous *GATA6* mutation along with an isogenic mutation-corrected control to be evaluated for cardiac directed differentiation. $GATA6^{1071delG/+}$ cells generated cTnT$^+$ CMs less efficiently than $GATA6^{corr/+}$ cells, on average about 50% (*Figure 1—figure supplement 1H*), indicating that the *GATA6* c.1071delG mutation similarly disrupts cardiac differentiation as observed in the hESC heterozygous mutant lines. Likewise, the CMs obtained by differentiation of either $GATA6^{1071delG/+}$ or $GATA6^{corr/+}$ iPSCs appeared morphologically normal (*Figure 1—figure supplement 1I*). The fact that both types of heterozygous cells (patient-derived iPSCs and targeted H1-hESCs) generate similar defects in CM differentiation provides evidence supporting the use of the H1-hESCs and isogenic lines to study the genetic and cellular basis for CHD. The corrected iPSC line was shown to have a grossly normal karyotype and to restore higher levels of GATA6 protein as expected (*Figure 1—figure supplement 1J and K*).

The impaired CM generation from *GATA6* mutant hESCs may be due to deficient CPC specification. Therefore, RT-qPCR assays were performed on $GATA6^{+/-}$, $GATA6^{-/-}$ and WT cells harvested at day 6 of cardiac differentiation to examine the expression levels of the CPC markers *NKX2.5*, *TBX5*, *TBX20*, and *GATA4*. CPC marker expression levels were significantly reduced in $GATA6^{+/-}$ and $GATA6^{-/-}$ cells relative to WT controls, indicating a defect in CPC specification (*Figure 1D*). At this time point there was a clear trend but not a statistically significant difference in the expression levels of CPC markers comparing the $GATA6^{+/-}$ and $GATA6^{-/-}$ genotypes. We also analyzed expression levels for markers of the first heart field (*HAND1* and *SMARCD3*) and the second heart field (SHF) (*ISL1*). All were significantly depleted in the $GATA6^{-/-}$ cells, while both *ISL1* and *SMARCD3* (but not *HAND1*) levels were significantly lower in the $GATA6^{+/-}$ cells. We note that an RT-qPCR time course showed that CPC marker expression level was increased (although not to WT levels) in $GATA6^{+/-}$ cells at later stages (days 8–11) while $GATA6^{-/-}$ CPC gene expression levels remained 'flat' in comparison with no increase over time (*Figure 1—figure supplement 2A*).

$GATA6^{+/+}$, $GATA6^{+/-}$, and $GATA6^{-/-}$ hESCs were differentiated toward the CM lineage using an alternative protocol based on chemical manipulation of the WNT pathway (*Lian et al., 2012*; CHIR protocol, *Figure 1—figure supplement 2B–E*) to confirm if the phenotype is consistent as observed using the cytokine-based protocol. In vitro cardiac differentiation protocols for hPSCs are known to sometimes require extensive optimization due to cell line variation and can require testing various cell seeding densities, media component concentrations, and batch variations for crucial reagents (*Laco et al., 2018*; *Pekkanen-Mattila et al., 2009*). The CHIR protocol was overall suboptimal for cardiac differentiation using the H1-hESCs due to substantial variation in CM differentiation efficiency and the occasional failure to generate CMs in WT H1-hESCs. We therefore committed to using the cytokine-based protocol (*Figure 1A*) for the bulk of experiments in this study because it proved to be a more consistent CM differentiation protocol for the H1-hESCs. Nevertheless, in successful experiments performed using the CHIR protocol we observed the same phenotype in $GATA6^{-/-}$ cells and $GATA6^{+/-}$ cells using the CHIR protocol as with the cytokine-based protocol (data not shown), consistent with a previous report (*Sharma et al., 2020*). Bulk RNA-seq was performed at day 5 using the CHIR protocol (*Figure 1—figure supplement 2C–E*). Principal component analysis (PCA) showed that the cells derived from each genotype were well separated from each other (*Figure 1—figure supplement 2C*). Differential gene expression and gene ontology (GO) analyses revealed that $GATA6^{-/-}$ cells had strongly reduced expression levels of cardiac development-related gene sets including 'OFT septum

morphogenesis' and 'heart development' compared to WT controls (*Figure 1—figure supplement 2D*). Furthermore, while *GATA6*[+/-] cells had relatively higher expression levels for many of these genes compared to *GATA6*[-/-] cells, numerous CPC regulatory genes showed decreased levels compared to WT controls (including *NKX2.5*, *TBX20*, *TBX5*, *MEF2C*, and *HAND2*; *Figure 1—figure supplement 2E*), confirming an impairment of CPC development in differentiating *GATA6* mutant hESCs. GO analysis of genes with increased expression levels in *GATA6*[-/-] cells relative to WT included gene sets such as 'nervous system development' (*Figure 1—figure supplement 2D*), suggesting a bias toward a non-cardiac genetic signature. Notably, gene sets related to retinoic acid (RA) signaling were significantly increased in GATA6[-/-] cells relative to WT (*Figure 1—figure supplement 2D*), and *ALDH1A2* expression (the main enzyme that generates RA) was increased in *GATA6*[-/-] cells relative to WT from days 4–6 using the cytokine-based protocol (*Figure 1—figure supplement 2F*), consistent with a previous study of cardiac differentiation using *GATA6* loss-of-function hiPSCs (*Sharma et al., 2020*). Proper RA signaling is crucial for cardiogenesis including OFT development and cardiac chamber morphogenesis (*Perl and Waxman, 2019*), and inhibiting excessive RA signaling in *GATA6* mutant hiPSCs was shown to partially rescue CPC marker expression (*Sharma et al., 2020*). Thus, upregulated RA signaling in *GATA6*[-/-] hESCs during the CPC developmental stage may partially underlie the CM differentiation defects.

Examining protein expression at multiple stages during cardiac differentiation revealed the earliest (and highest) expression level for GATA6 at day 2 of differentiation (*Figure 2A*). The impaired CPC development in *GATA6*[+/-] and *GATA6*[-/-] cells may therefore be due to defective cardiac mesoderm patterning during the early stages of cardiogenesis. We examined co-expression of the precardiac mesoderm markers KDR and PDGFRα (KP) (*Kattman et al., 2011*) using flow cytometry from days 3–5 of cardiac differentiation (*Figure 2B*). At day 3, there was no significant difference in the %K[+]P[+] cells between genotypes (~45% K[+]P[+]), but by day 5 the %K[+]P[+] increased to ~72% in WT and *GATA6*[+/-] cells, while the %K[+]P[+] was markedly reduced in *GATA6*[-/-] cells (~22%, *Figure 2B and C*). This is consistent with previous reports where sorting and reseeding K[+]P[+] cells at day 3 of cardiac differentiation generated both mesenchymal cells and CMs but day 5 sorted K[+]P[+] cells gave rise predominantly to CMs, indicating that day 5 KP expression is relatively enriched for precardiac mesoderm (*Kattman et al., 2011*; *Takeda et al., 2018*). *GATA6*[+/-] cells had a small but significant decrease in the %K[+]P[+] cells at day 4 relative to WT (*Figure 2—figure supplement 1A and B*), suggesting that the developing K[+]P[+] cardiac mesoderm in *GATA6*[+/-] cells is temporarily defective or delayed relative to WT controls. Notably, the decreased %K[+]P[+] observed at day 4 (*GATA6*[+/-] and *GATA6*[-/-] cells) and day 5 (*GATA6*[-/-] cells) relative to WT cells was due to reduced KDR but not PDGFRα expression (*Figure 2D and E*, *Figure 2—figure supplement 1A and B*), indicating that KDR is (directly or indirectly) regulated by GATA6 during cardiac mesoderm patterning.

We next sought to determine whether GATA6 loss-of-function impaired mesoderm induction by analyzing the expression of the pan-mesoderm marker BRACHYURY (*T*). Flow cytometry performed on WT, *GATA6*[+/-], or *GATA6*[-/-] cells at day 2 or 3 of cardiac differentiation revealed no significant difference in abundance of BRACHYURY[+] cells between genotypes (*Figure 2F and G*). Likewise, there was no significant change in transcript levels for *T* or the mesendoderm marker *EOMES*, or mesoderm markers MESP1, or MESP2 at day 2 when assessed by RT-qPCR (*Figure 2H*). These results indicate that *GATA6* mutant hESCs have the capacity to differentiate into a nascent mesendoderm/early mesoderm population that is deficient in patterning cardiac mesoderm.

## Transcriptome analysis of *GATA6* mutant hESCs during early mesoderm patterning

During gastrulation, sequentially lineage-restrictive mesodermal cell populations emerge along a developmental trajectory involving primitive streak to lateral mesoderm formation, followed by cardiac mesoderm patterning (*Miquerol and Kelly, 2013*). As *GATA6* loss-of-function hESCs differentiated to an emergent mesendoderm/mesoderm stage but failed to properly generate cardiac mesoderm, we hypothesized that GATA6 is required during the earliest stages of lateral and precardiac mesoderm patterning. GATA6-regulated genes important during these early mesoderm patterning stages were investigated by performing bulk RNA-seq at day 2 or 3 of cardiac differentiation (*Figure 3—figure supplement 1A and B*). Again, PCA showed the profiles representing each genotype were well separated. GO analysis and gene set enrichment analysis (GSEA) using the biological process

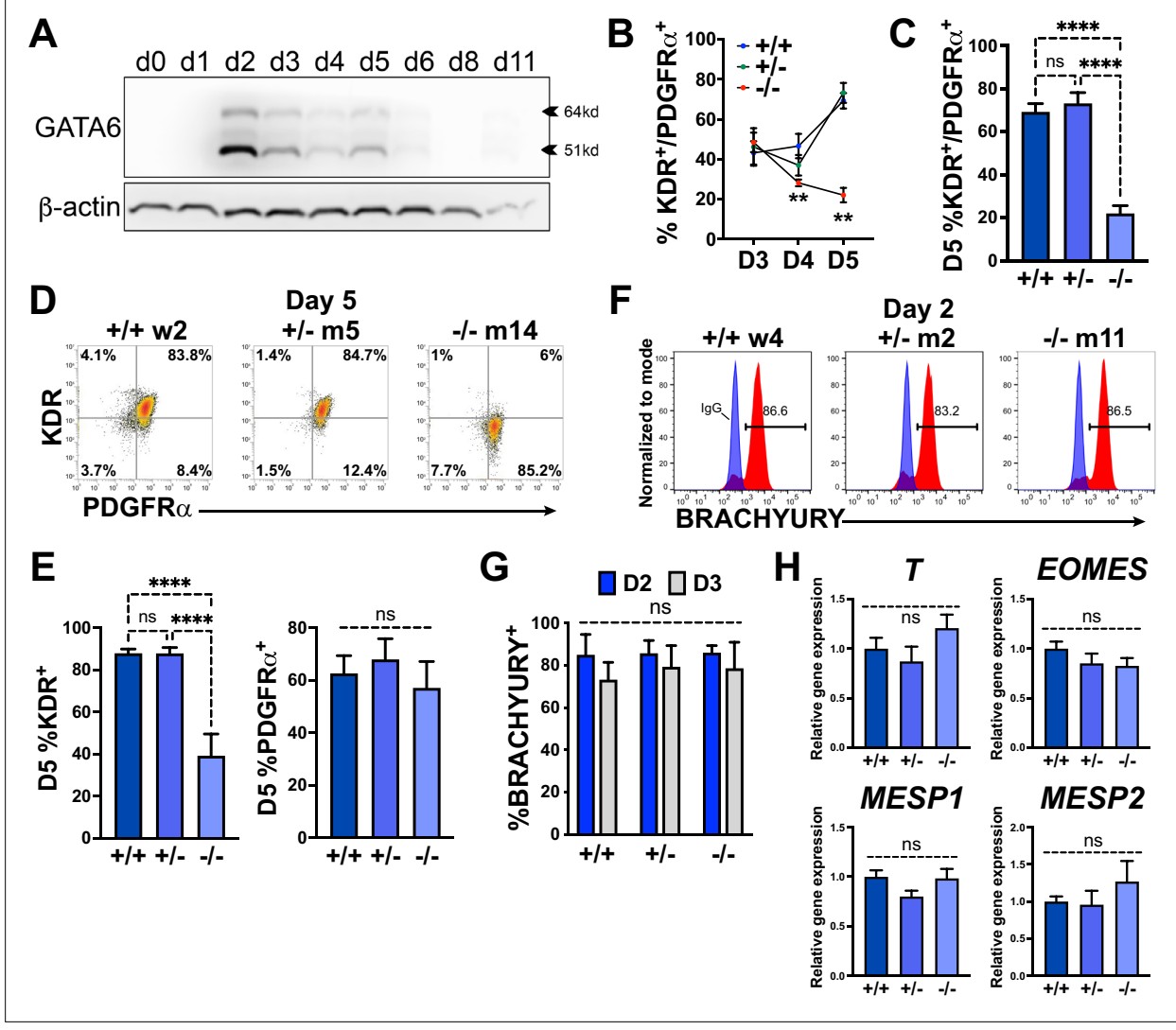

**Figure 2.** GATA6 is required for cardiac mesoderm development. (**A**) Western blot time course using protein lysates from *GATA6+/+* human embryonic stem cells (hESCs) probed for GATA6 with β-actin used as a loading control. (**B**) Flow cytometry quantification for % KDR and PDGFRα double-positive (%K+P+) cells from days 3–5 of cardiac differentiation of *GATA6+/+*, *GATA6+/-*, or *GATA6-/-* hESCs (n ≥ 4). Asterisks indicate statistical significance comparing *GATA6-/-* and WT on day 4 or 5 of cardiac differentiation. (**C**) Day 5 flow cytometry quantification for %K+P+ cells (n = 7). (**D**) Representative flow cytometry plots for day 5 %K+P+ cells. (**E**) Day 5 flow cytometry quantification (n = 7) for %KDR+ (left) or PDGFRα+ (right). (**F**) Representative flow cytometry plots for day 2 %BRACHYURY+ cells (red) overlaid IgG stained controls (blue). (**G**) Quantification for day 2 or day 3 %BRACHYURY+ cells (n = 4). (**H**) RT-qPCR for day 2 *T, EOMES, MESP1*, and *MESP2* expression levels normalized to *GATA6+/+* samples (n = 6). Data represents the mean ± SEM, with significance indicated as **p<0.01, ****p<0.0001, and ns indicating not significant by two-way ANOVA (**B, G**) or one-way ANOVA (**C, E, H**) with Tukey's multiple-comparison test. The labels w4, w2 (wildtype), m2, m5 (heterozygous), and m11, m14 (homozygous) refer to the isogenic wildtype and mutant hESC clones (see *Figure 1—figure supplement 1B*).

The online version of this article includes the following source data and figure supplement(s) for figure 2:

**Source data 1.** Western blots in panel A.

**Source data 2.** Flow cytometry in panels B, C, E and G.

**Source data 3.** RT-qPCR in panel H.

**Figure supplement 1.** KDR and PDGFRα cardiac mesoderm analysis at day 4.

**Figure supplement 1—source data 1.** KP flow cytometry in panel B.

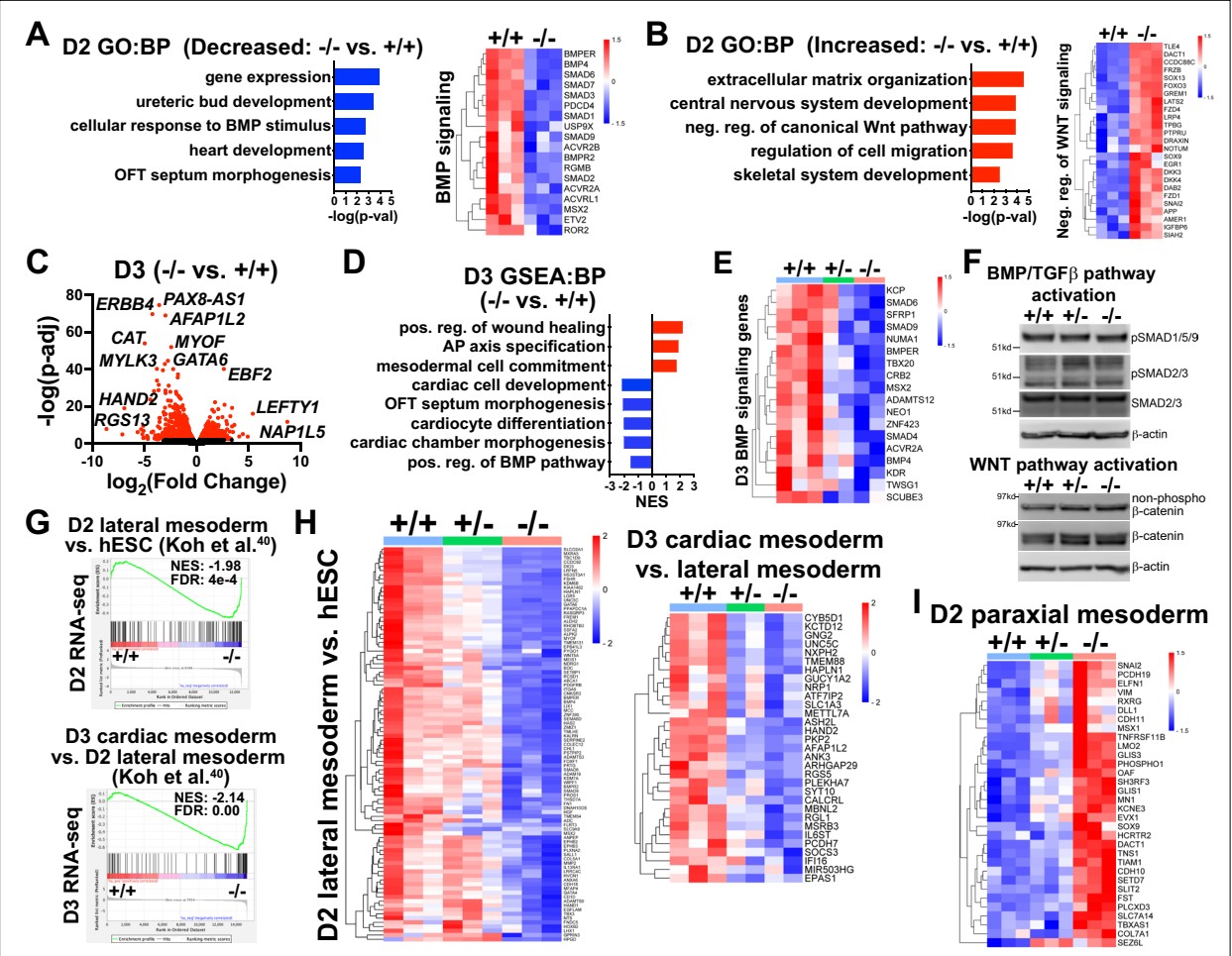

**Figure 3.** Transcriptome analysis at early mesoderm patterning stages. (**A**) Gene ontology (GO) analysis for biological process (BP) from decreased differentially expressed genes (dDEGs) identified comparing $GATA6^{-/-}$ to WT samples from day 2 RNA-seq data (left). Heatmap for genes related to BMP signaling from day 2 RNA-seq shown on the right. Color gradient on heatmap indicates relative gene expression levels. (**B**) GO analysis (BP) of increased DEGs identified comparing $GATA6^{+/-}$ to WT samples from day 2 RNA-seq (left). Heatmap for genes related to negative regulation of canonical WNT pathway from day 2 RNA-seq shown on the right. (**C**) Volcano plot for day 2 $GATA6^{-/-}$ sample RNA-seq gene expression data relative to WT controls. Dots represent genes, red indicates p-adj<0.05 and black indicates p-adj>0.05. (**D**) Gene set enrichment analysis (GSEA) (BP) of $GATA6^{-/-}$ cells relative to WT controls from day 3 RNA-seq data. NES indicates normalized enrichment score. (**E**) Heatmap for BMP signaling genes from day 3 RNA-seq data. (**F**) Western blots using protein lysates from $GATA6^{+/+}$, $GATA6^{+/-}$ or $GATA6^{-/-}$ cells at day 2 of cardiac differentiation probed with antibodies recognizing phospho-SMAD1/5/9, phospho-SMAD2/3, total SMAD2/3, non-phospho-β-catenin, and total β-catenin with β-actin used as a loading control. (**G**) GSEA analysis of day 2 or 3 RNA-seq gene expression data ($GATA6^{-/-}$ relative to WT) using the day 2 lateral mesoderm (relative to human embryonic stem cell [hESC], left) and day 3 cardiac mesoderm (relative to day 2 lateral mesoderm, right) during hESC cardiac differentiation datasets from *Koh et al., 2016*. FDR indicates false discovery rate. (**H**) Heatmaps from RNA-seq day 2 (left) or 3 (right) data using core enrichment genes identified in (**G**) for lateral mesoderm (relative to hESCs) and cardiac mesoderm (relative to lateral mesoderm). (**I**) Heatmap showing enriched paraxial mesoderm gene expression in the mutant cells.

The online version of this article includes the following source data and figure supplement(s) for figure 3:

**Source data 1.** Western blots in panel F.

**Source data 2.** Original unmarked western blot images in panel F.

**Figure supplement 1.** RNA-seq analysis following cardiac differentiation at day 2 or 3.

(BP) subcollection for differentially expressed genes (DEGs; defined as p-adj<0.05 and log₂(fold change)>0.5) comparing $GATA6^{-/-}$ to WT samples revealed decreased enrichment for GO terms including 'heart development' (e.g., *HAND1*) and 'OFT septum morphogenesis' (e.g., *NRP1*) at days 2 and 3, suggesting a disruption of early cardiac signature genes (*Figure 3A, C and D*). The decreased enrichment for the 'OFT septum morphogenesis' GO term is consistent with the conclusions of a

previous study of *GATA6* mutations in hPSCs in the context of cardiac differentiation (*Sharma et al., 2020*); however, the results presented here implicate this program at a much earlier developmental stage (D2 mesendoderm/lateral mesoderm). Conversely, GO analysis of increased DEGs in *GATA6*⁻/⁻ samples relative to WT at days 2 and 3 included 'extracellular matrix (ECM) organization', 'skeletal system development', and 'AP axis specification' (*Figure 3B and D*), suggesting that *GATA6*⁻/⁻ cells are biased to other lineages such as paraxial mesoderm or retain an early mesodermal phenotype by day 3.

*GATA6*⁺/⁻ samples also had reduced expression levels for genes associated with cardiac GO terms such as 'cardiac muscle contraction' and 'OFT septum morphogenesis' when comparing to WT controls at day 3 (*Figure 3—figure supplement 1C and D*). Indeed, there was overlap for many of the decreased DEGs (dDEGs) observed in *GATA6*⁺/⁻ and *GATA6*⁻/⁻ cells at day 2 or day 3 (relative to WT) including cardiac development genes such as *ERBB4* (important for endocardial cushion and ventricular trabeculae development *Gassmann et al., 1995*), the SHF-related transcription factor *HAND2*, and the cardiac mesoderm marker *LRRC32* (*Loh et al., 2016*), although the decrease in expression levels for these genes relative to WT was not as severe in the *GATA6*⁺/⁻ genotype as for *GATA6*⁻/⁻ (*Figure 3—figure supplement 1D–F*).

We observed dysregulation of genes related to the WNT and BMP networks in *GATA6*⁻/⁻ samples relative to WT at day 2 (*Figure 3—figure supplement 1G*) and GO and GSEA revealed decreased levels of 'positive regulation of the BMP pathway' and increased levels of 'negative regulation of canonical WNT pathway' at day 2 or 3 relative to WT (*Figure 3A, B, D and E*). Several dDEGs identified in *GATA6*⁻/⁻ cells at day 2 or 3 relative to WT samples included negative regulators of WNT signaling, including *SFRP1* and *TLE1* (*Figure 3—figure supplement 1E and G*), and antagonists of the BMP pathway, such as *BMPER* and *SMAD6* (*Figure 3A and E*, *Figure 3—figure supplement 1E and G*), indicating that the overall transcriptional response induced by the WNT and BMP pathways is dysregulated. *GATA6*⁺/⁻ cells also exhibited perturbed gene expression for several WNT- and BMP-related genes (at an intermediate level to those observed in *GATA6*⁻/⁻ cells) at day 2 or 3 relative to WT (*Figure 3E*, *Figure 3—figure supplement 1G*). Mesoderm patterning is controlled by the input of WNT, BMP, and TGFβ signaling and dysfunction in these pathways cause severe developmental defects (*Prummel et al., 2020*; *Kimelman and Griffin, 2000*). Therefore, the cardiac defects caused by GATA6 loss-of-function may be caused by these alterations in WNT and BMP signaling.

We examined activity of the BMP, TGFβ, and WNT pathways by analyzing the effector proteins SMAD1/5/9 (BMP activated), SMAD2/3 (TGFβ activated), and β-catenin (WNT activated) by western blotting at day 2. There was no change in the levels of active phospho-SMAD1/5/9, phospho-SMAD2/3, or non-phospho-β-catenin when comparing *GATA6*⁺/⁻, *GATA6*⁻/⁻, and WT cells (*Figure 3F*). These results indicate that GATA6 regulates the transcriptional response of genes activated by the BMP and WNT pathways downstream of canonical effector protein activation.

To further examine the impact of *GATA6* loss-of-function on early mesoderm patterning, we correlated RNA-seq data from a study examining hESC differentiation toward cardiac mesoderm (*Koh et al., 2016*) to our day 2 or 3 RNA-seq data. GSEA using increased DEGs at day 2 (lateral mesoderm) or 3 (precardiac mesoderm) from *Koh et al., 2016* revealed significantly reduced enrichment in our *GATA6*⁻/⁻ samples (relative to WT) on day 2 or 3, respectively (*Figure 3G and H*). Furthermore, *GATA6*⁺/⁻ cells had partially reduced expression for these lateral and precardiac mesoderm genes relative to WT cells (*Figure 3H*), suggesting that the impaired CPC and CM differentiation observed for *GATA6*⁺/⁻ hESCs originates as disrupted lateral and cardiac mesoderm patterning. We additionally compared the increased DEGs at day 2 from Koh et al. for paraxial mesoderm and our day 2 RNA-seq data and observed increased expression for these genes in *GATA6*⁻/⁻ cells relative to controls (*Figure 3I*), further supporting the notion that the *GATA6*⁻/⁻ hESCs are at biased toward a paraxial mesoderm-like genetic signature.

A previous study of GATA6 function during DE differentiation from hESCs reported an interaction between GATA6, EOMES, and SMAD2/3 to co-regulate genes important during mesendoderm and transition to DE development (*Chia et al., 2019*). DE and cardiac mesoderm originate from a bipotent germlayer mesendoderm population (*Tada et al., 2005*) and likely share a gene expression signature at this early stage. We therefore performed GSEA using the CHIP-seq dataset for triple-overlap of GATA6, EOMES, and SMAD2/3 during early DE differentiation from *Chia et al., 2019* and found a significantly decreased enrichment in *GATA6*⁻/⁻ cells relative to WT at day 2 for putative direct GATA6

target genes (*Figure 3—figure supplement 1H*). EOMES is an important transcriptional regulator of anterior mesoderm development (*Probst et al., 2021*), and as GATA6 was shown to interact with EOMES during early DE differentiation (*Chia et al., 2019*; *Heslop et al., 2022*) we hypothesized that an interaction is similarly necessary during early lateral mesoderm patterning. We therefore performed GSEA using another published dataset of EOMES targets identified during mesoderm and DE (ME) differentiation in mouse ESCs (mESCs) (*Tosic et al., 2019*). GSEA of our day 2 RNA-seq data with the 'EOMES ME direct activation dataset' human gene equivalents revealed a significant decrease in enrichment in *GATA6[-/-]* cells relative to WT (*Figure 3—figure supplement 1H*), supporting the notion that GATA6 cooperates with EOMES to co-regulate genes required for lateral mesoderm patterning.

## GATA6 binds to putative regulatory regions associated with BMP, WNT, and EOMES-related dDEGs

To determine the genomic localization of GATA6 occupancy during mesoderm patterning, we performed cleavage under targets and release using nuclease (CUT&RUN) sequencing for GATA6 at day 2 of differentiation using GATA6[+/+] hESCs. 1273 significant differentially bound sites were identified for GATA6 relative to IgG control samples, associated with 953 transcriptional start sites (TSS) for unique genes (*Figure 4A*). Significant GATA6 binding sites predominately localized to intronic (46.03%) and distal intergenic (32.21%, >5 kb from TSS) regions (*Figure 4A*), consistent with previous reports of GATA6 binding distribution (*Sulahian et al., 2014*; *Fisher et al., 2017*). Transcription factor binding motif enrichment analysis revealed the strongest enrichment at GATA6 bound loci for 'GATA' motifs, as well as 'LHX-like' (LIM homeobox transcription factor; expressed in cardiac and other mesodermal derivatives; *Costello et al., 2015*) and 'EOMES' motifs (*Figure 4B*). GO analysis of the gene list nearby significant GATA6 binding peaks showed striking similarity to GO analysis from RNA-seq dDEGs in *GATA6[-/-]* cells (relative to WT), including 'OFT septum morphogenesis', 'BMP signaling pathway', and 'WNT signaling pathway' (*Figure 4C*). Additionally, we observed an overlap of CUT&RUN identified genes with day 2 (17.9%) and 3 (16.8%) RNA-seq dDEGs in *GATA6[-/-]* cells (relative to WT) representing high likelihood direct targets of GATA6 including the cardiac development-related genes *HAND1* and *MYOCD* (*Figure 4D*). Visualizing the localization of these GATA6 peaks that mapped to day 2 and 3 dDEGs confirmed binding in intronic regions or at distal intergenic regions that overlapped with known enhancer-like signatures identified by the ENCODE Project (*Moore et al., 2020*; *Figure 4E*). These included several WNT-related genes such as *MCC* and *LGR5*, as well as BMP-related genes including *SMAD6* (*Figure 4E*). In *Figure 4F*, genes are highlighted in red that have GATA6 binding peaks in associated putative enhancers that are likely to be direct targets for the WNT/BMP networks. These are distinguished from WNT/BMP pathway genes that were not bound by GATA6 yet are downregulated in the GATA6 mutant cells and are more likely to be indirect targets.

Day 2 GATA6 CUT&RUN data revealed an enrichment for binding to sequences containing EOMES motifs (*Figure 4B*). We compared our day 2 GATA6 CUT&RUN and RNA-seq data to previously published EOMES CHIP-seq data performed on hESCs at day 2 of DE differentiation (*Teo et al., 2011*). Comparing these datasets revealed 32 genes that had triple-overlap of day 2 GATA6 CUT&RUN identified genes (953 genes nearby significantly bound sites), day 2 RNA-seq dDEGs, and EOMES CHIP-seq targets (*Teo et al., 2011*; *Figure 4G*). Among this list of triple-overlap genes are known cardiac development factors including *LGR5*, which was reported to be required for hESC-CM differentiation (*Jha et al., 2017*), and *PRDM1*, which is expressed in the SHF and involved in OFT morphogenesis (*Vincent et al., 2014*). Visualizing the location of these triple-overlap genes confirmed that the GATA6 peaks directly overlapped with many EOMES CHIP-seq peaks (*Figure 4—figure supplement 1*), suggesting that GATA6 and EOMES interact to co-regulate genes required for precardiac mesoderm patterning.

## GATA6 interactome analysis during precardiac to cardiac mesoderm patterning stages

We next analyzed the GATA6 interactome using rapid immunoprecipitation of endogenous proteins (RIME) on WT cells at day 2 or 4 of cardiac differentiation to define the chromatin associated protein interactions. 99 significant interacting proteins at day 2 and 52 unique proteins at day 4 of cardiac differentiation were identified bound to GATA6 compared to IgG control samples (*Figure 5A*). EOMES was one of the most enriched proteins bound to GATA6 at day 2 but was not identified in the day 4 GATA6-RIME analysis (*Figure 5B*). 81.8% of proteins identified by GATA6-RIME at day 2

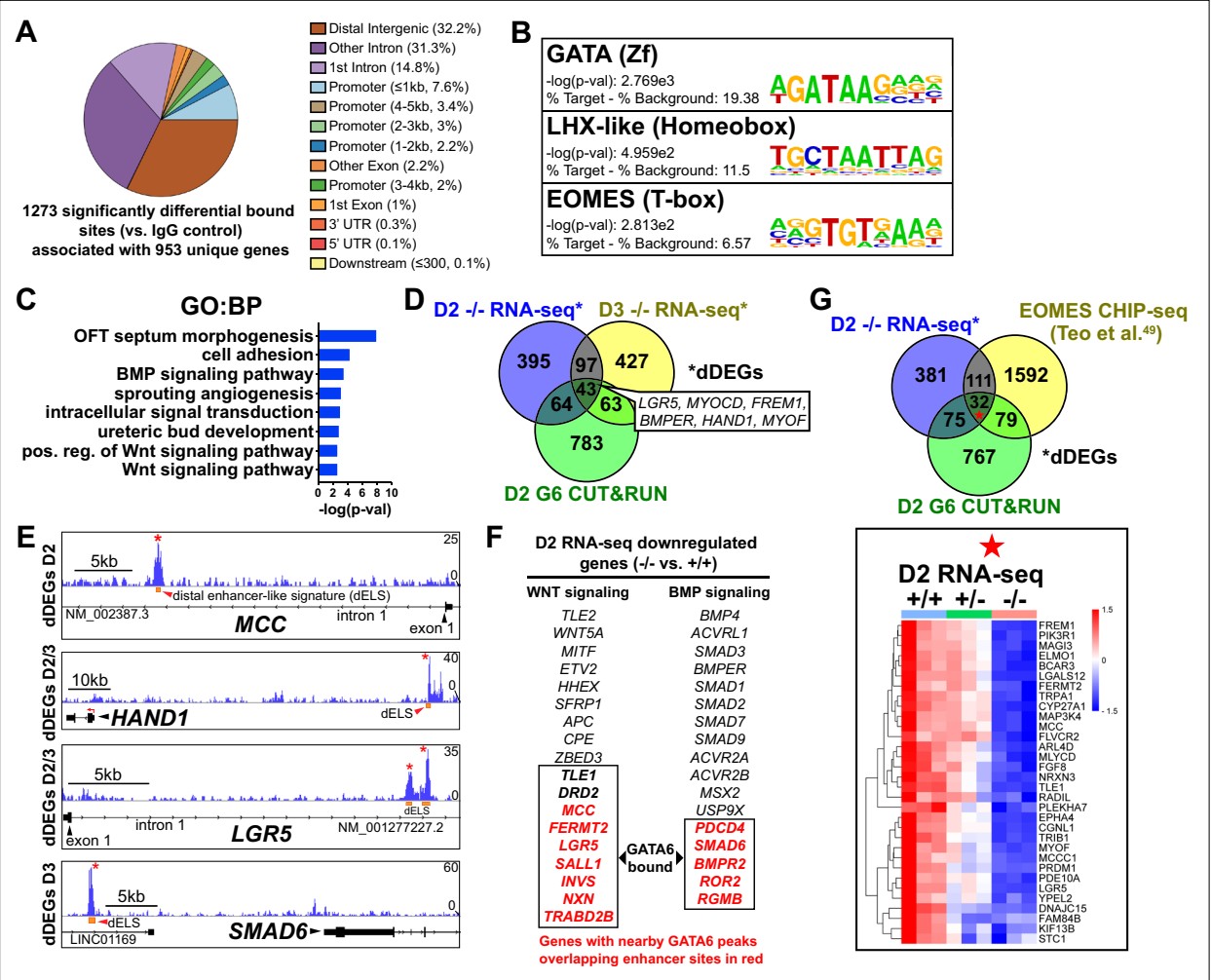

**Figure 4.** GATA6 CUT&RUN analysis during early mesoderm patterning. (**A**) Genomic distribution for significant GATA6 binding peaks identified by GATA6 CUT&RUN at day 2 of cardiac differentiation (n = 3). (**B**) Transcription factor motif enrichment at GATA6 bound loci. (**C**) Gene ontology (GO) (biological process [BP]) analysis of the gene list associated with significant GATA6 binding peaks. (**D**) Venn diagram comparisons for day 2 GATA6 CUT&RUN identified genes (green), day 2 decreased differentially expressed genes (dDEGs) (blue), and day 3 dDEGs (yellow) identified by RNA-seq (*GATA6*-/- relative to WT). (**E**) Human genome browser representations of GATA6 CUT&RUN data (blue tracks) aligned to select genes that are dDEGs identified by day 2 or 3 RNA-seq (*GATA6*-/- relative to WT). Orange rectangles represent approximate location for distal enhancer-like signatures (dELS) via the ENCODE Project. Asterisks indicate significant GATA6 binding peaks (p<0.003 relative to IgG controls). (**F**) Genes from the WNT and BMP signaling pathways that are significantly downregulated in the mutant cells compared to wildtype. Genes in red were found by CUT&RUN to have GATA6 binding peaks in associated putative enhancers and are therefore likely to be direct targets. (**G**) Venn diagram comparisons for day 2 GATA6 CUT&RUN identified genes (green), day 2 dDEGs (blue, *GATA6*-/- relative to WT), and EOMES-CHIP-seq identified genes at day 2 of hESC-DE differentiation from *Teo et al., 2011* (yellow). Inlayed heatmap indicates *GATA6*+/+, *GATA6*+/- and *GATA6*-/- RNA-seq gene expression data at day 2 of cardiac differentiation for the 32 triple-overlap genes.

The online version of this article includes the following figure supplement(s) for figure 4:

**Figure supplement 1.** GATA6 CUT&RUN peaks overlap with previously published EOMES CHIP-seq peaks.

were not encoded by significantly DEGs identified by day 2 RNA-seq (*GATA6*-/- vs. WT) while 18.2% were, including TLE1, VRTN, and ZIC2 (*Figure 5—figure supplement 1A*), indicating that GATA6 functions in parallel with most interacting proteins identified. Many of the significantly bound proteins were components of chromatin remodeling complexes such as members of the SWI/SNF complex (e.g., SMARCC1, SMARCA4, and ARID1A), which functions to mobilize nucleosomes and promote DNA accessibility (*Mittal and Roberts, 2020*), and the Nucleosome Remodeling and Deacetylase (NuRD) complex (e.g., KDM1A, MTA1/2/3, and GATAD2A), a chromatin remodeling complex associated with transcriptional repression (*Millard et al., 2016*; *Figure 5B*). GATA6 protein interactions were

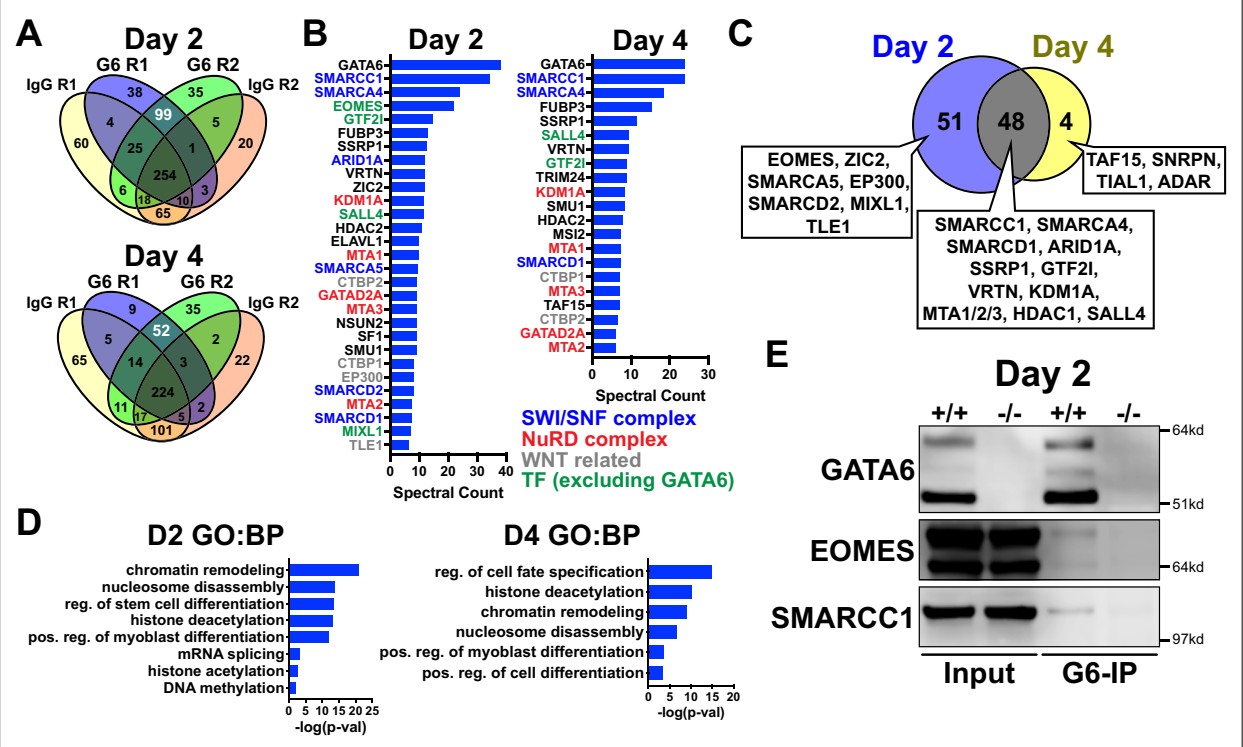

**Figure 5.** GATA6 interactome analysis during precardiac to cardiac mesoderm patterning stages. (**A**) Venn diagrams showing unique proteins identified by RIME analysis performed on *GATA6+/+* hESCs at day 2 or 4 of cardiac differentiation. Numbers in white text indicate enriched proteins identified by GATA6-RIME. G6 indicates GATA6, R indicates replicate (n = 2). (**B**) Enriched proteins identified by GATA6 RIME (spectral count >5, unique peptides >1). (**C**) Venn diagram comparing day 2 (blue) and 4 (yellow) GATA6-RIME-enriched proteins. (**D**) Gene ontology (GO) (biological process [BP]) analysis for GATA6-RIME-enriched proteins on day 2 or 4. (**E**) Western blots for GATA6, EOMES, and SMARCC1 performed on GATA6-immunoprecipitated (G6-IP) whole-cell protein lysates from *GATA6+/+* and *GATA6-/-* cells isolated at day 2 of cardiac differentiation. Input indicates whole-cell protein lysate controls.

The online version of this article includes the following source data and figure supplement(s) for figure 5:

**Source data 1.** Western blots (GATA6-IP) in panel E.

**Source data 2.** Original unmarked western blot images in panel E.

**Figure supplement 1.** Extended GATA6-RIME analysis.

also observed for other epigenetic and transcriptional regulators including those related to histone deacetylation (i.e., HDAC2) and mRNA splicing (i.e., SF1, *Figure 5B and D*). Interestingly, WNT/β-catenin transcriptional regulators were identified bound to GATA6, particularly at day 2 of cardiac differentiation, including the β-catenin transcriptional co-activator EP300 (*Sun et al., 2000*), and the TCF/LEF transcriptional co-repressors CTBP1/2 (*Valenta et al., 2003*; *Cuilliere-Dartigues et al., 2006*) and TLE1 (*Brantjes et al., 2001*; *Figure 5B*), suggesting that GATA6 may directly modulate the transcriptional response induced by WNT signaling by binding to these proteins.

Proteins identified by GATA6-RIME only at day 2 included the transcription factors EOMES and MIXL1 (both transcription factors involved in primitive streak formation and mesoderm patterning *Loh et al., 2016*; *Stavish et al., 2020*) as well at TLE1, EP300, and SMARCA5, while the unique day 4 proteins were all RNA binding proteins such as TAF15 (*Figure 5C*). The overlapping day 2 and 4 GATA6-RIME results included many chromatin remodelers such as SMARCC1, SMARCA4, and KDM1A (*Figure 5C*), although the enrichment for these proteins was greater at day 2 compared to day 4 (*Figure 5B*, *Supplementary file 1*). As day 2 of differentiation had the highest level of GATA6 expression and enrichment for RIME identified proteins, we performed GATA6 immunoprecipitation (IP) on *GATA6+/+* and *GATA6-/-* samples at day 2 and confirmed a weak interaction with EOMES and SMARCC1 in the WT samples, with no observable bands in *GATA6-/-* samples (*Figure 5E*). The EOMES and SMARCC1 protein bands identified in GATA6-IP samples were notably less than observed for

the input controls; therefore, it is possible that the interaction between GATA6 and these proteins is indirect (i.e., GATA6 may be co-bound in a complex with EOMES and/or SMARCC1 among other proteins).

Previous studies performed GATA6-RIME at day 4 and EOMES-RIME at day 2 of DE differentiation (*Heslop et al., 2021*; *Heslop et al., 2022*), so we compared our day 2 and 4 GATA6-RIME data to these previously published RIME datasets (*Figure 5—figure supplement 1B*). 26 proteins were present in all four datasets (*Figure 5—figure supplement 1B and C*), implying there is a common interaction between GATA6 and various remodeler proteins, the transcription factor SALL4, and the TCF/LEF co-repressors CTBP1/2 during mesendoderm patterning toward both cardiac mesoderm and DE lineages. Protein interactions unique to day 2 GATA6- and EOMES-RIME included MIXL1, SMARCC2, and ARID1B and may represent protein interactions specific to mesendoderm development (*Figure 5—figure supplement 1C*). Notably, EOMES and the WNT-related proteins TLE1 and EP300 were not identified in our day 4 dataset but were present in our day 2 GATA6-RIME data and the published EOMES and GATA6-RIME datasets during early DE development (*Heslop et al., 2021*; *Heslop et al., 2022*). Therefore, an interaction between GATA6 and these proteins appears to be relinquished by the cardiac mesoderm stage (day 4) but persists during DE patterning. Cardiac differentiation requires a transient inhibition of the canonical WNT signaling pathway (*Ueno et al., 2007*), and as day 4 of our cardiac differentiation protocol occurs after WNT inhibition these varying protein interactions may reflect the differing contributions of the WNT pathway during cardiac and DE differentiation.

## Early manipulation of the WNT and BMP pathways partially rescues the CM defects in GATA6 mutant hESCs

*LGR5*, a component of the WNT pathway (*de Lau et al., 2014*), was among the most significant dDEGs identified from day 2 or 3 RNA-seq in GATA6$^{-/-}$ cells (relative to WT) and was also found significantly bound by GATA6 (and in a previous study, EOMES; *Teo et al., 2011*) in WT cells (*Figure 4—figure supplement 1*). As LGR5 is required for efficient CM differentiation in hPSCs (*Jha et al., 2017*; *Sahara et al., 2019*), we hypothesized that infection with a doxycycline (DOX)-inducible *LGR5*-expressing lentivirus might rescue the cardiac defects in *GATA6*$^{-/-}$ hESCs. In *GATA6*$^{-/-}$ hESCs that had been transduced with the inducible-*LGR5* lentivirus (i*LGR5*), DOX was added from days 1–4 of cardiac differentiation using the cytokine-based protocol (*Figure 6A*) to mimic the *LGR5* expression pattern (*Figure 6—figure supplement 1A*); we validated that this increased *LGR5* expression in a DOX-dose-dependent manner (*Figure 6—figure supplement 1B*). In all differentiation experiments using i*LGR5* treated with DOX (250 ng/mL), we did not observe generation of beating CMs. However, DOX treatment of i*LGR5*-transduced *GATA6*$^{-/-}$ hESCs did yield a small but significant increase in the %K$^+$P$^+$ population at day 5 of differentiation relative to empty vector (EV) transduced controls, associated with increased KDR expression (*Figure 6B*). Thus, *LGR5* expression can partially rescue a proportion of day 5 cardiac mesoderm in *GATA6*$^{-/-}$ hESCs but cannot sufficiently improve subsequent CM differentiation.

As LGR5 is a single component of the complex WNT pathway, we tested a broader pharmacological approach by using the WNT agonist CHIR to determine if early WNT activation could more efficiently rescue the cardiac defects in *GATA6*$^{-/-}$ hESCs. We found that inclusion of CHIR (3 µM) from days 0–2 in differentiating *GATA6*$^{-/-}$ hESCs yielded by day 13 small clusters of beating CMs (*Video 1*). When assessing quantitatively by flow cytometry the %cTnT$^+$ CMs in CHIR-treated *GATA6*$^{-/-}$ cells relative to vehicle-treated controls, there was a variable trend for CM induction that was however not statistically significant (*Figure 6—figure supplement 1C*). Similarly, while the %K$^+$P$^+$ cardiac mesoderm cells at day 5 was slightly increased upon CHIR treatment in *GATA6*$^{-/-}$ cells, this was not a statistically significant improvement (*Figure 6—figure supplement 1D*).

BMP-related gene expression was also dysregulated at day 2 in *GATA6*$^{-/-}$ cells and thus manipulation of the BMP4 concentration in combination with CHIR treatment from days 0–2 was tested to further promote CM rescue. Interestingly, we found that decreasing the BMP4 concentration (5 ng/mL) combined with CHIR treatment yielded small clusters of beating CMs (*Video 2*) more consistently than increasing the BMP4 concentration (data not shown) or treatment with CHIR on its own. Quantification by flow cytometry for %cTnT$^+$ CMs in *GATA6*$^{-/-}$ cells treated with CHIR and the lower BMP4 concentration (CHIR LB) showed a small but significant increase (~4% cTnT$^+$) relative to *GATA6*$^{-/-}$ controls indicating that CM differentiation is partially rescued upon CHIR LB treatment (*Figure 6C*).

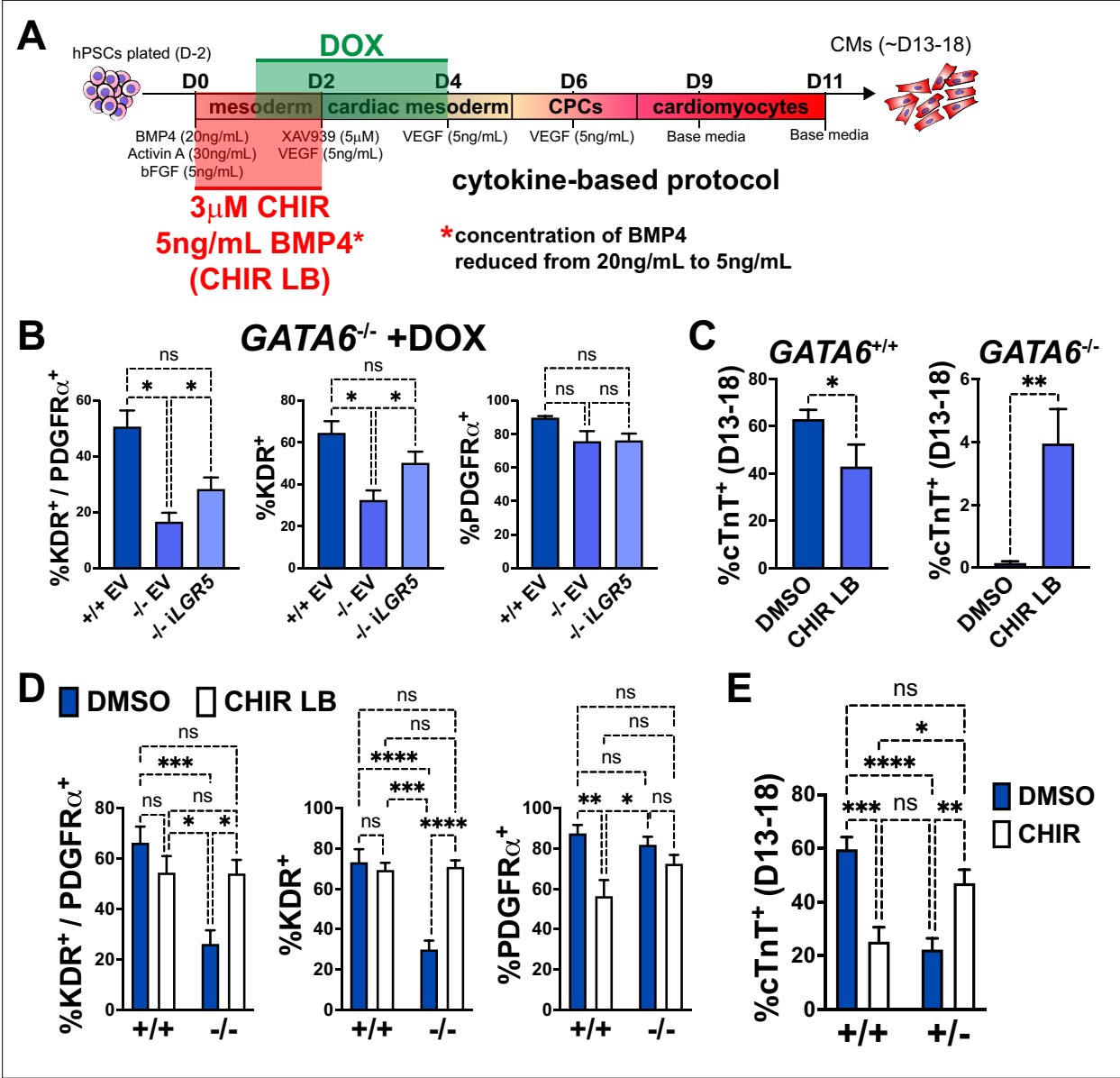

**Figure 6.** Early manipulation of the WNT and BMP pathways partially rescues the cardiomyocyte (CM) defects in GATA6 loss-of-function human embryonic stem cells (hESCs). (**A**) Schematic for treatment with DOX (days 1–4), CHIR (3 μM, days 0–2), and/or reduced BMP4 (5 ng/mL, days 0–2, indicated as 'LB') during CM directed differentiation using the cytokine-based protocol. (**B**) Day 5 flow cytometry quantification for %K⁺P⁺ double-positive cells, %KDR⁺ single-positive cells, or %PDGFRα⁺ single-positive cells in WT or *GATA6⁻/⁻* hESCs transduced with i*LGR5* or empty vector (EV) (n = 5). Significance indicated by *p<0.05 according to one-way ANOVA and Tukey's post hoc analysis. (**C**) %cTnT⁺ CMs from days 13–18 of cardiac differentiation quantified by flow cytometry in WT or *GATA6⁻/⁻* cells treated with CHIR LB or vehicle (DMSO) with normal BMP4 concentration treated control (n ≥ 6). (**D**) Flow cytometry at day 5 of cardiac differentiation to quantify %K⁺P⁺ double-positive, %KDR⁺ single-positive, or %PDGFRα⁺ single-positive cells comparing WT or *GATA6⁻/⁻* hESCs treated with CHIR LB with *GATA6⁺/⁺* and *GATA6⁻/⁻* hESCs controls treated with vehicle and normal BMP4 concentration (n ≥ 8). (**E**) %cTnT⁺ CMs from days 13–18 of cardiac differentiation quantified by flow cytometry in *GATA6⁺/⁻* or WT hESCs treated with CHIR (3 μM) or DMSO (n ≥ 7). Data represents the mean ± SEM, significance indicated by **p<0.01, ***p<0.001, ****p<0.0001 by two-tailed Student's *t*-test (**C**) and two-way ANOVA (**D, E**) with Tukey's multiple-comparison test.

The online version of this article includes the following source data and figure supplement(s) for figure 6:

**Source data 1.** Flow cytometry in panels B-E.

**Figure supplement 1.** Extended data for i*LGR5* and early CHIR treatment.

**Figure supplement 1—source data 1.** RT-qPCR in panels A and B.

**Figure supplement 1—source data 2.** cTnT and KP flow cytometry in panels C and D.

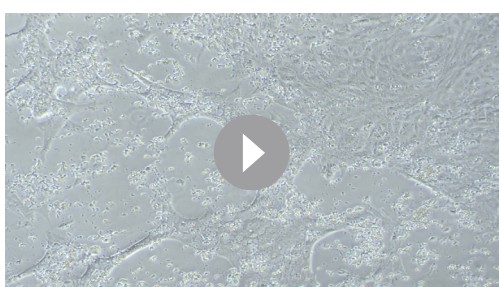

**Video 1.** Inclusion of CHIR (3 µM) from days 0–2 in differentiating *GATA6*⁻/⁻ human embryonic stem cells (hESCs) yielded by day 13 small clusters of beating cardiomyocytes (CMs). Shown is a representative cluster at day 16, ×10 magnification.

https://elifesciences.org/articles/100797/figures#video1

Note that this treatment actually decreases efficiency for differentiation in the control WT cells. The %K⁺P⁺ and %K⁺ day 5 cardiac mesoderm population was also significantly increased with CHIR LB treatment in *GATA6*⁻/⁻ cells relative to untreated controls and was not significantly different than WT controls indicating a rescue of day 5 K⁺P⁺ cardiac mesoderm (*Figure 6D*). Taken together, these results indicate that modulating the early input of the WNT and BMP pathways in *GATA6*⁻/⁻ cells can rescue the cardiac mesoderm patterning defects and partially rescue subsequent CM differentiation.

*GATA6*⁺/⁻ cells also exhibited partial dysregulated gene expression related to WNT signaling via day 2 RNA-seq relative to WT controls (*Figure 3—figure supplement 1G*). We therefore hypothesized that the addition of CHIR from days 0–2 in differentiating *GATA6*⁺/⁻ hESCs could rescue CM differentiation more potently than observed in *GATA6*⁻/⁻ hESCs. Indeed, we found that the %cTnT⁺ CMs was significantly increased in *GATA6*⁺/⁻ cells supplemented with CHIR relative to vehicle-treated controls and was not significantly different from WT untreated controls, indicating an efficient rescue of CM differentiation (*Figure 6E*). CHIR addition to WT hESCs had a deleterious effect on CM differentiation efficiency, with the %cTnT⁺ CMs reduced from ~60% (DMSO treated) to ~25% (CHIR treated, *Figure 6E*), suggesting that there is a threshold 'window' for the degree of WNT pathway activation required during early primitive streak to mesoderm patterning stages and that reducing or exceeding this threshold has an adverse effect on later CM differentiation capacity. Furthermore, this data indicates that the rescue in CM differentiation efficiency in *GATA6*⁺/⁻ cells treated with CHIR was not due to an overall improvement in the general cardiac differentiation protocol, but rather related to correcting the consequence of loss of one *GATA6* allele.

## Discussion

We found that *GATA6* loss-of-function mutations severely disrupt CPC gene expression and impair CM differentiation, consistent with the results described by *Sharma et al., 2020*. Here, we discovered that GATA6 is required very early in this process and that the later cardiac phenotypes observed in *GATA6*⁺/⁻ and *GATA6*⁻/⁻ hESCs originate from an early mesoderm patterning defect in which lateral and cardiac mesoderm genes fail to be expressed at normal levels. Gene networks induced by the WNT and BMP pathways that are normally required for primitive streak organization and subsequent anterior lateral mesoderm emergence are also mis-expressed when GATA6 is deficient. GATA6 CUT&RUN analysis revealed binding at distal enhancer-like signatures nearby dDEGs identified by RNA-seq, and GATA6 binding was found to have a considerable overlap with previously published EOMES CHIP-seq data during early DE differentiation (*Teo et al., 2011*) nearby many of these genes. Day 2 GATA6 RIME analysis confirmed binding with EOMES and various chromatin remodelers including components of the SWI/SNF and NuRD remodeling complexes that are known to regulate chromatin accessibility of cardiac lineage-specific gene networks (*Lei et al., 2012*; *Hota et al., 2019*; *Hota et al., 2022*; *Herchenröther et al., 2023*; *Shi et al., 2024*). Furthermore, we found that early manipulation of the WNT pathway through CHIR treatment from

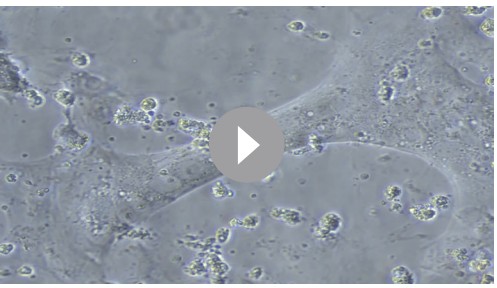

**Video 2.** Decreasing the BMP4 concentration (5 ng/mL) combined with CHIR treatment yielded small clusters of beating cardiomyocytes (CMs) from GATA6-/- cells more consistently than increasing the BMP4 concentration or treatment with CHIR on its own. Shown is a representative cluster at day 18, ×40 magnification.

https://elifesciences.org/articles/100797/figures#video2

days 0–2 rescued CM defects observed in *GATA6*[+/-] hESCs while CHIR treatment and reduced BMP4 dosage rescued K[+]P[+] cardiac mesoderm and partially rescued CM differentiation in *GATA6*[-/-] hESCs. Together, these results present multiple novel functions for GATA6 in regulating early precardiac mesoderm patterning during human cardiac differentiation.

WNT and BMP-related genes were dysregulated in *GATA6*[+/-] and *GATA6*[-/-] cells at day 2 and 3 of cardiac differentiation but β-catenin and SMAD protein activation was intact indicating that the transcriptional response of these pathways is defective rather than the upstream activation induced by morphogens. GATA6 RIME analysis revealed an interaction with several WNT transcriptional regulators including the β-catenin co-activator EP300 and the TCF/LEF transcriptional co-repressors TLE1 and CTBP1/2, and binding to these proteins could have positive or negative effects on their function. One potential mechanism is that GATA6 binds to these WNT regulators and prevents their repressive (or in the case for EP300, potentiating) function on TCF-mediated transcription, causing a net increase in the β-catenin transcriptional response. Indeed, there was an enrichment for negative regulation of the WNT signaling pathway at day 2 in *GATA6*[-/-] cells according to RNA-seq data, suggesting that GATA6 loss-of-function increases feedback inhibition of TCF/LEF-mediated transcription. GATA6 CUT&RUN analysis also revealed binding near many genes related to WNT and BMP signaling implying direct participation in modulating appropriate β-catenin/SMAD transcriptional responses. However, many WNT/BMP response genes dysregulated in *GATA6*[+/-] and *GATA6*[-/-] cells were not identified as targets by GATA6 CUT&RUN, suggesting that another method for regulation is indirect, perhaps related to the interaction with chromatin remodelers affecting DNA accessibility.

RIME analysis revealed GATA6 bound to many chromatin remodelers including components of the SWI/SNF (e.g., SMARCA4) and NuRD (e.g., KDM1A) complexes at days 2 and 4 of cardiac differentiation. GATA6 is a pioneer transcription factor with specialized function to bind and promote accessibility of chromatin during CPC development (*Sharma et al., 2020*), and GATA6, GATA4, and GATA3 have previously been shown to interact with various members of SWI/SNF complexes such as SMARCA4 (*Sharma et al., 2020*; *Heslop et al., 2021*; *Takaku et al., 2016*; *Lickert et al., 2004*). However, the molecular mechanism for how GATA6 modifies the accessibility of chromatin and mobilizes nucleosomes is unclear. The AP-1 pioneer factor interacts with SWI/SNF complexes to promote their localization to non-permissive regions of chromatin (*Wolf et al., 2023*). GATA6 may function similarly, for example, by binding to SWI/SNF complex proteins and recruiting them to regions of closed chromatin containing lineage-specific enhancers where they then function to promote genomic accessibility. It is somewhat surprising that among all the SWI/SNF factors, the one that is known to be cardiac-specific (SMARCD3) was not found in the GATA6-RIME-enriched proteins. We note that SMARCD3 is expressed in the RNA-seq data as early as day 2. Although speculative, it may be that GATA6 primarily interacts with other SWI/SNF complexes prior to a subsequent role for SMARCD3 in cardiac specification.

The function of GATA6 during early mesoderm patterning presented in this study has unique implications toward understanding the pathogenesis of CHD and extra-cardiac phenotypes associated with human *GATA6* mutations. *GATA6* homozygous loss-of-function mutations have not been described in humans, most likely due to early embryonic lethality causing non-viable embryogenesis. This is consistent with the early mesoderm patterning defects of *GATA6*[-/-] hESCs described in this study, previous reports of DE defects in *GATA6*-null hPSC lines (*Shi et al., 2017*; *Heslop et al., 2021*; *Chia et al., 2019*; *Tiyaboonchai et al., 2017*), as well as the early lethality reported in *GATA6* KO mice (*Morrisey et al., 1998*). In the case of haploinsufficiency, CHD presenting as OFT or septal defects associated with heterozygous *GATA6* mutations may be due to defects that occur during early mesoderm patterning that diminish the later developmental pool of CPCs and CMs. Quite remarkably, profiles of cells at day 2 of differentiation already identify OFT morphogenesis as gene sets disturbed by loss of GATA6. Relatively minor changes in early mesoderm cell population numbers have recently been reported to associate with later heart defects in a mouse model of Cornelia de Lange syndrome using *NPBL1*[+/-] mice (*Chea et al., 2024*), and thus *GATA6* heterozygous mutations may similarly have a subtle negative effect on developing mesodermal cell populations to cause CHD. Human mutations in *GATA6* commonly associate with OFT defects, a primarily SHF-derived structure, and recent work has established that SHF lineage progenitors are derived from early defined mesodermal subpopulations (*Yang et al., 2022*). Therefore, *GATA6* mutations might impair specific heart field progenitor populations based on the timing and positioning of these developing mesodermal populations in the

primitive streak, long before the progenitors are associated with SHF. *GATA6* haploinsufficiency in humans also associates with endoderm defects such as pancreatic agenesis, and DE and early meso-derm co-develop from a common mesendoderm population adjacent to one another in the anterior primitive streak during gastrulation (*Murry and Keller, 2008*). Defects in the regulatory functions of GATA6 during mesendoderm patterning, such as interactions with EOMES and specific chromatin remodelers, may cause downstream organ defects in both mesoderm and endoderm-derived organ systems in some patients with *GATA6* mutations. The data presented here focuses on early stages of mesoderm patterning and cardiac progenitor differentiation in the context of *GATA6* loss-of-function rather than CM profiling and is thus a limitation of our study. Future investigation of how heterozygous *GATA6* mutations impact CMs physiology such as cardiac chamber identity and CM function would be a worthwhile pursuit to expand our knowledge of how *GATA6* regulates CM differentiation in relation to human CHD.

## Materials and methods

### hPSC culture

hPSC lines were cultured on Matrigel (Corning BD354277)-coated tissue culture plates using StemFlex (Thermo Fisher Scientific A3349401) or mTESR Plus (Stem Cell Technologies 1000276) medium at 37°C and 5% $CO_2$. Cells were passaged using Accutase Cell Detachment Solution (VWR 10761-312) and supplemented with 10 μM ROCK inhibitor (RI) Y-27632 (Selleckchem S1049) for 1 day. hPSCs were routinely tested for mycoplasma contamination using the LookOut Mycoplasma PCR Detection Kit (Sigma-Aldrich MP0035).

### iPSC generation

A skin biopsy was obtained from a patient with an atrial septal defect and congenital diaphragmatic hernia associated with a heterozygous frameshift mutation in *GATA6* (c.1071delG) (*Yu et al., 2014*). Human dermal fibroblasts (HDFs) derived from this biopsy were transduced with four Sendai viruses expressing OCT3/4, SOX2, KLF4, and c-MYC using the CytoTune-iPS 2.0 Sendai Reprogramming Kit (Invitrogen A16517). Transduced HDFs were reprogrammed on CF1 mouse embryonic fibroblasts (MEFs) (Thermo Fisher Scientific A34180) with iPSC medium consisting of DMEM/F12 (VWR 16777-255) supplemented with 20% KnockOut serum replacement (KOSR, Thermo Fisher Scientific 10828010), 1% non-essential amino acids (Thermo Fisher Scientific), 1% L-glutamine (Corning 25-005Cl), 1% Peni-cillin:Streptomycin Solution (Corning 30-002Cl), 0.0007% 2-mercaptoethanol (Thermo Fisher Scientific 21985023), 10 ng/mL bFGF (R&D Systems 233-FB-025). Once iPSC colonies emerged, colonies were manually isolated and expanded into feeder-free conditions by culturing with mTESR1 (Stem Cell Technologies 85850) on Matrigel coated plates.

### CRISPR gene editing

The GATA6 mutant H1 (WA01, NIH #0043) hESC cell lines were established in a previous study (*Shi et al., 2017*) using the H1-iCas9 cell line and CRISPR gene editing (*González et al., 2014*) targeting the C-terminal ZnF domain (*Figure 1—figure supplement 1A and B*). The *GATA6* c.1071delG iPSC mutant allele was corrected to WT sequence to create the isogenic *GATA6*^corr/+ line (as well as unmod-ified *GATA6*^1071delG+ CRISPR-control clones) using CRISPR technology with Nickase-Cas9 according to previously established methods (*Ran et al., 2013*). A pair of gRNA sequences were designed targeting opposing DNA strands surrounding the G deletion site. The gRNA 1 sequence was specif-ically designed to overlap the G deletion site to prevent Nickase-Cas9 activity on the WT *GATA6* allele. A 91 nucleotide (nt) length single-stranded DNA (ssDNA) repair template (Integrated DNA Technologies) was designed with 45 nt homology arms flanking the G mutation site, including a G to A silent mutation in the right homology arm and gRNA 2 recognition sequence to prevent Nickase-Cas9 activity after homology-directed repair (HDR). gRNAs 1 and 2 (*Supplementary file 2*) were cloned into the pSpCas9n(BB)–2A-Puro (PX462) V2.0 plasmid (Addgene 62987) as described previously (*Ran et al., 2013*). 2 million dissociated *GATA6* c.1071delG/+iPSCs were electroporated with 4 μg each gRNA-PX462 vector with 3 μL of 100 μmol/L ssDNA donor oligonucleotide (see *Supplementary file 2*) using the Neon Transfection System (two pulses at 1450 V for 20 ms, Invitrogen NEON1S) and plated on Matrigel coated dishes with mTESR Plus supplemented with 10 μM RI. A T7 endonuclease

I assay (New England Biolabs M0302S) was performed to quantify indel frequency of gRNA pairs following PCR amplification (Phusion Hot Start Flex DNA Polymerase, New England Biolabs M0535L) and agarose gel electrophoresis from genomic DNA isolated 2–3 days after electroporation using the DNeasy Blood & Tissue Kit (QIAGEN 69506). 24 hr after electroporation, the culture medium was replaced with mTESR Plus supplemented with 0.5 μg/mL puromycin (Sigma-Aldrich P8833) for antibiotic resistance selection for 48 hr. iPSCs were then dissociated with Accutase and plated at a low density (~3000 iPSCs per 10 cm plate) on Matrigel coated dishes with mTESR Plus medium (with 10 μM RI for the first 24 hr) and allowed to grow for about 2 weeks before iPSC colonies were manually isolated and plated into Matrigel coated 96-well plates and cultured with mTESR Plus. DNA was isolated from iPSC clones using QuickExtract DNA Extraction Solution (Biosearch Technologies QE09050), PCR amplification was performed using primers surrounding the mutation site (*Supplementary file 2*), and Sanger sequencing was performed to verify gene editing (GENEWIZ). Karyotyping was performed by the WiCell Research Institute (Madison, WI).

## hPSC cardiomyocyte differentiation

A monolayer cytokine-based cardiac differentiation protocol was adapted from previously established methods (*Figure 1A*; *Dubois et al., 2011*; *Kattman et al., 2011*). Briefly, a 'base medium' for cardiac differentiation was prepared using RPMI 1640 (Thermo Fisher Scientific 22400089) supplemented with 0.5× B27 (Thermo Fisher Scientific 17504044), 0.5 mM 1-Thioglycerol (Sigma-Aldrich M6145), 50 μg/mL L-ascorbic acid (Sigma-Aldrich A4544), and 1× GlutaMAX (Thermo Fisher Scientific 35050061). The hPSCs were plated at ~4 × 10$^4$ cells/cm$^2$ and allowed to grow for 2 days before changing with 'Day 0 medium' consisting of the 'base medium' plus 20 ng/mL BMP4 (R&D Systems 314BP-050), 30 ng/mL Activin A (R&D Systems 338-AC-050), 5 ng/mL bFGF (R&D Systems 233-FB-025), and 5 μL/mL Transferrin (Roche 10652202001). BMP4 was used at 10 ng/mL for iPSC differentiation. Approximately 55 hr later, the medium was changed with 'Day 2 medium' consisting of 'base medium' supplemented with 5 μM XAV-939 (Sigma-Aldrich X3004), 5 ng/mL VEGF (R&D Systems 293-VE-050), and 5 μL/mL Transferrin. The culture medium was changed on days 4 and 6 of cardiac differentiation with 'base medium' supplemented with 5 ng/mL VEGF. From day 9 and later, the culture medium was changed with 'base medium' every 2–3 days. For a subset of experiments, 3 μM CHIR99021 (Stem Cell Technologies 72054) was included in the 'Day 0 medium' and/or the concentration of BMP4 was reduced to 5 ng/mL (*Figure 6C and E*, *Figure 6—figure supplement 1C and D*). An alternative CM differentiation protocol based on chemical manipulation of the WNT pathway (*Lian et al., 2012*) was used for a subset of experiments (CHIR protocol, *Figure 1—figure supplement 2B–E*). For the CHIR protocol, hESCs were plated at ~4 × 10$^4$ cells/cm$^2$ and allowed to grow for 2 days in mTESR Plus before changing medium to RPMI 1640 with 1× B27 minus insulin (Thermo Fisher Scientific A1895601) and 6 μM CHIR99021 (Stem Cell Technologies 72054) for 24 hr. Medium was then changed to RPMI 1640 with 1× B27 minus insulin for another 24 hr, before changing medium on day 3 to RPMI 1640 with 1× B27 minus insulin and 5 mM IWP2 (Tocris 3533). On day 5, culture medium was changed to RPMI 1640 with 1× B27 minus insulin once more and starting on day 7 medium was changed to RPMI with 1× B27 every other day. CMs were purified using a lactate-selection protocol to prepare for immunocytochemistry by changing the differentiation medium on day 15 of the cytokine-based protocol to a lactate-selection medium consisting of RPMI 1640 lacking glucose (Thermo Fisher Scientific 11879020) supplemented with 5 mM sodium L-lactate (Sigma-Aldrich L7022), 213 μg/mL L-ascorbic acid, and 500 μg/mL Human Albumin (Sigma-Aldrich A9731). Lactate-selection medium was refreshed on day 17 of differentiation, after which the medium was restored to 'base medium' on day 19 and changed every 2–3 days prior to fixation.

## Flow cytometry

Cells were detached from culture plates using 0.5% Trypsin/0.5 mM EDTA (VWR 45000-664) or TrypLE Express Enzyme (Thermo Fisher Scientific 12605010) and resuspended in a FACS buffer consisting of PBS with 10% FBS (for cTnT staining) or 0.5% BSA (for live cell staining). Cells were fixed by pelleting at 300 × *g* and resuspended in 0.2% paraformaldehyde (PFA) for 25 min at room temperature (RT) before staining with primary antibody for cTNT (Thermo Fisher Scientific MA5-12960) in FACS buffer with 0.5% Saponin (Sigma-Aldrich S7900-25G) for 1 hr at RT, washed 3× in FACS buffer, and then stained for 1 hr at RT with FACS buffer plus Saponin containing secondary antibody (Invitrogen A21236). Live

cells were used for staining the cell surface proteins KP by incubating with FACS buffer containing PE-conjugated KDR antibody (R&D Systems FAB357P-100) and APC-conjugated PDGFRα antibody (R&D Systems FAB1264A) or IgG control antibodies (R&D Systems IC002P, IC002A) for 30 min at RT, and then stained with SYTOX Green Dead Cell Stain (Thermo Fisher Scientific S34860) in FACS buffer for 5 min at RT. BRACHYURY flow cytometry was performed using the Foxp3 Transcription Factor Staining Buffer Set (eBioscience 00-5523-00) according to the manufacturer's protocol using a PE-conjugated BRACHYURY antibody (R&D Systems IC2085P) or IgG control antibody (R&D Systems IC108P). Flow cytometry was performed and analyzed using an Attune NxT Flow Cytometer (Thermo Fisher Scientific) for fixed cells and a BD-Accuri C6 Flow Cytometer (BD Biosciences) for live cells.

## RT-qPCR

RNA was isolated using the RNeasy Mini Kit (QIAGEN 74106) and reverse-transcribed using the SuperScript VILO cDNA Synthesis Kit (Invitrogen 11754250) according to the manufacturer's protocols. The qPCR reaction was prepared by mixing cDNA with LightCycler 480 SYBR Green Master mix (Roche 04887-352-001) and primers listed in *Supplementary file 2*. RT-qPCR was performed using a LightCycler 480 Instrument II (Roche). Gene expression data was normalized to endogenous *HPRT* expression levels as a control for each sample.

## Western blotting

Whole-cell lysates were prepared using RIPA Lysis Buffer (Thermo Fisher Scientific 89900) supplemented with 1× Protease/Phosphatase Inhibitor Cocktail (Cell Signaling Technology 5872S) and 1× Benzonase (Millipore 70664-3). Protein was quantified using the Pierce BCA Protein Assay Kit (Thermo Fisher Scientific 23225) and electrophoresed on NuPAGE 4 to 12% Bis-Tris Gels (Invitrogen NP0335BOX) before transferring to PVDF membranes (Bio-Rad 162-0177). Blots were washed with 1× TBS with 0.1% Tween-20 (TBST) and blocked using 5% IgG-free BSA (VWR RLBSA50). Primary antibodies (*Supplementary file 2*) were added to blocking buffer and stained overnight at 4°C. Secondary antibody staining was performed for 1 hr at RT using HRP-conjugated antibodies (*Supplementary file 2*) in blocking buffer. Pierce ECL Western Blotting Substrate (Thermo Fisher Scientific 32106) used to visualize proteins, and imaging was performed using a C-DiGit Blot Scanner (LI-COR) with ImageStudio (v5.2) and Adobe Photoshop (v25.9.1) software.

## Co-immunoprecipitation

Co-immunoprecipitation (co-IP) was performed using the Pierce Crosslink Magnetic IP/Co-IP Kit (Thermo Fisher Scientific 88805) according to the manufacturer's protocol using cells at day 2 of cardiac differentiation. 7 µg of the GATA6 antibody (Cell Signaling Technology 5851) was used to cross-link to the Protein A/G Magnetic beads and 1 mg of protein was used for the IP reaction performed overnight at 4°C on a rotating platform. GATA6 Co-IP samples were electrophoresed on a gel and imaged as described for western blotting using 30 µg of protein lysate from the Co-IP used as an input control. *GATA6*−/− cell samples served as a negative control.

## Immunocytochemistry and microscopy

Cells were fixed in tissue culture dishes with 4% PFA for 20 min at RT. Cells were washed in PBS and then blocked for 1 hr at RT with blocking buffer consisting of PBS with 10% FBS, 0.1% IgG-free BSA (Rockland Immunochemicals BSA-10), 0.1% Saponin, and 10% goat serum (Jackson ImmunoResearch Laboratories Inc 005-000-121). Cells were then stained with primary antibodies (*Supplementary file 2*) diluted in blocking buffer (without goat serum) overnight at 4°C. Secondary antibody staining was performed in blocking buffer (without goat serum) for 1 hr at RT. Cells were stained with DAPI to visualize nuclei and imaged using a Zeiss LSM800 laser scanning confocal microscope with ZEN imaging software or a Zeiss Epifluorescence microscope with AxioVision software, and edited using Adobe Photoshop software (v25.9.1). Videos were taken using an Infinity5 Microscope Camera (Teledyne Lumenera) with Infinity Analyze software.

## Lentivirus production and infection

*LGR5* full-length cDNA was first cloned into a DOX-inducible Lentiviral vector containing a GFP selection marker (CS-TRE-PRE-Ubc-tTA-I2G; *Yamaguchi et al., 2012*) by performing PCR for LGR5 using

WT cDNA samples from day 2 of cardiac differentiation and primers designed to incorporate the BsiWI and MfeI restriction enzyme recognition sites flanking LGR5 (*Supplementary file 2*) to allow digestion and insertion into the CS-TRE-PRE-Ubc-tTA-I2G vector. *LGR5* sequence was validated by Sanger sequencing (GENEWIZ) using sequencing primers listed in *Supplementary file 2*. Lentivirus production was performed by transfecting 293T cells with the lentiviral expression vectors (LGR5 or EV control) and packaging plasmids using PEI. Transfected 293T cells were then allowed to produce lentivirus for 48 hr before the medium was collected, and the virus was concentrated using Lenti-X Concentrator (Takara 631231) according to the manufacturer's protocol and resuspended in StemFlex medium. *GATA6*$^{-/-}$ hESCs were infected with lentivirus by replacing medium with i*LGR5* or EV control (*Lan et al., 2021*) lentivirus concentrate in StemFlex for 16 hr before changing to normal StemFlex and seeded for 2–3 days. Lentivirus-infected hESCs were then sorted based on GFP expression using FACS through the Weill Cornell Medicine Flow Cytometry Core Facility. Doxycline (DOX, Sigma-Aldrich D9891) was added from days 1–4 to induce expression of i*LGR5* or EV infected *GATA6*$^{-/-}$ hESCs.

## RNA-seq

Total RNA was extracted from hESCs approximately 52 hr (day 2), 74 hr (day 3), or 5 days after starting the cardiac differentiation protocol using the RNeasy Mini Kit (QIAGEN 74106) from at least two independent biological replicates per sample. Total RNA integrity was verified using a 2100 Bioanalyzer (Agilent Technologies), and RNA concentrations were measured using the NanoDrop system (Thermo Fisher Scientific). Libraries were generated from days 2 and 5 RNA using the TruSeq RNA Library Prep Kit v2 (Illumina RS-122-2001) and day 3 RNA using the TruSeq Stranded mRNA Library Prep Kit (Illumina 20020594) according to the manufacturer's instructions. Normalized cDNA libraries were pooled and sequenced on Illumina HiSeq4000 (days 2 and 5, single read) or Illumina NovaSeq 6000 (day 3, paired-end read) sequencing systems for 50 cycles through the Weill Cornell Medicine Genomics Resources Core Facility. The raw sequencing reads in BCL format were processed through bcl2fastq 2.19 (Illumina) for FASTQ conversion and demultiplexing. Day 2 and 5 RNA-seq alignment (GRChg19) and differential gene expression analysis was performed as described (*Anelli et al., 2017*). Day 3 RNA-seq reads were aligned and mapped to the GRCh38 human reference genome using STAR (version 2.5.2; *Dobin et al., 2013*) and transcriptome resconstruction was performed using Cufflinks (version 2.1.1, http://cole-trapnell-lab.github.io/cufflinks/) after trimming adaptors with cutadapt (version 1.18, https://cutadapt.readthedocs.io/en/v1.18/). The abundance of transcripts was measured with Cufflinks in Fragments Per Kilobase of exon model per Million mapped reads (FPKM) (*Trapnell et al., 2013*; *Trapnell et al., 2010*). Raw read counts per gene were extracted using HTSeq-count v0.11.2 (*Anders et al., 2015*). Gene expression profiles were constructed for differential expression, cluster, and PCA with the DESeq2 package (*Love et al., 2014*). For differential expression analysis, pairwise comparisons between two or more groups using parametric tests where read-counts follow a negative binomial distribution with a gene-specific dispersion parameter were performed. Corrected p-values were calculated based on the Benjamini–Hochberg method to adjust for multiple testing. GSEA was performed using GSEA 4.2.3 software (Broad Institute, Inc and Regents of the University of California) (*Subramanian et al., 2005*) on RNA-seq gene lists ranked by log$_2$(Fold Change) using the subcollection of biological process (BP) gene sets from the GO collection in the Human Molecular Signatures Database (MSigDB) (*Castanza et al., 2023*), or using custom gene lists from published sequencing datasets as described in this article (*Chia et al., 2019*; *Koh et al., 2016*; *Tosic et al., 2019*), and gene sets with a false discovery rate (FDR) *q*-value < 0.25 and a nominal p-value<0.05 were considered to be enriched significantly. GO analysis was performed using the Database for Annotation, Visualization, and Integrated Discovery (DAVID, david.ncifcrf.gov; *Huang et al., 2009*; *Sherman et al., 2022*) using the BP subcollection. Heatmaps were generated from normalized counts from selected gene lists using SRplot (bioinformatics.com.cn/srplot; *Tang et al., 2023*). DEGs were defined as p-value adjusted (p-adj)<0.05 and log$_2$(fold change)>0.5, genes with a count below 50 for all samples were excluded.

## CUT&RUN

500,000 H1-*GATA6*$^{+/+}$ cells were harvested approximately 52 hr after starting the cardiac differentiation protocol, and CUT&RUN was performed using the CUTANA ChIC/CUT&RUN Kit (Epicypher 14-1048) according to the manufacturer's protocol. Anti-GATA6 (n = 3 independent biological replicates) (Cell

Signaling Technology 5851) and rabbit anti-IgG (n = 2 independent biological replicates) negative control (Epicypher 13-0042) antibodies were used at a 1:50 dilution (~0.5 µg). CUT&RUN library preparation, sequencing, and SEACR-based analysis were performed by the Weill Cornell Medicine Epigenomics and Genomics Core Facilities. The DNA libraries were prepared with approximately 5 ng of CUT&RUN enriched DNA using the NEBNext Ultra II DNA Library Prep Kit for Illumina (New England Biolabs E7645) and NEBNext Multiplex Oligos for Illumina (96 Unique Dual Index Primer Pairs) (New England Biolabs E6440); 14 PCR cycles were used to amplify the adaptor ligated target DNA. The libraries were clustered on a pair end flow cell and sequenced for 50 cycles on an Illumina NextSeq500 Sequencing System. FASTQ files were aligned to the GENCODE GRCh38 build of the human genome using bowtie2 with the –-dovetail parameter. The resulting BAM files were sorted and indexed using samtools and then converted to bedgraph format using bedtools. Peaks were subsequently called using SEACR to identify enrichment in target data by selecting the top 1% of regions by AUC (stringent threshold). The DiffBind R Bioconductor package was used to interrogate differential binding for GATA6 versus IgG controls, and the annotated significant nearby TSS gene list from the SEACR analysis was used for genomic peak distribution, GO analysis (as described for RNA-seq), and Venn diagram comparisons. Transcription factor motif enrichment analysis was performed using Basepair software with a CUT&RUN Peaks, Motif (MACS2, HOMER) pipeline (basepairtech.com). EOMES CHIP-seq data from *Teo et al., 2011* was aligned to GRChg38 using bowtie2 with Partek software (partek.com). CUT&RUN and CHIP-seq data were visualized using Integrative Genomics Viewer (IGV) 2.16.2, and distal enhancer-like signatures (dELS) defined by the ENCODE Project (*Moore et al., 2020*) were identified using the UCSC Genome Browser for hg38 (genome.ucsc.edu).

## RIME analysis

About ~50 million H1-*GATA6*$^{+/+}$ cells at day 2 or 4 of cardiac differentiation were fixed with 1% formaldehyde (Electron Microscopy Sciences 15710) for 8 min at RT with gentle agitation. Fixation was quenched with 125 mM glycine (Bio-Rad 1610718) for 5 min at RT before cells were manually scraped, transferred to conical tubes, and pelleted at $800 \times g$ at 4°C for 10 min. Pellets were resuspended in PBS with 0.5% IGEPAL CA-630 (Sigma-Aldrich I8896) and centrifuged again before resuspending in PBS-IGEPAL with 1 mM PMSF (Thermo Fisher Scientific 36978) and centrifuged one final time. Supernatant was then removed from pellets which were then snap frozen on dry ice and transferred to –80°C. RIME analysis was performed by Active Motif (Carlsbad, CA) according to published methods (*Mohammed et al., 2016*) using 150 µg of chromatin per sample immunoprecipitating with anti-GATA6 (Cell Signaling Technology 5851) or anti-IgG control (Cell Signaling Technology 2729) antibodies (15 µg antibody per IP) and performing mass spectrometry with enriched GATA6 bound proteins defined as not present in the IgG controls and having a spectral count (the total number of all peptides detected) of at least 5 and >1 unique peptides.

## Statistical analysis

Data is represented as the mean ± SEM from independent biological replicates (as indicated in figure legends). Graphing and statistical analysis were performed using GraphPad Prism v.10.2.3 software with two-tailed Student's *t*-test when comparing two groups. One-way or two-way ANOVA was performed when comparing three or more groups, unless otherwise indicated.

## Acknowledgements

JAB, DH, and TE conceived and planned the study. JAB generated most of the data, generated figures, and wrote the first manuscript draft. MG and KMB provided significant assistance and consult on experimental strategies and bioinformatics. RK and ZS generated and validated the iPSC lines. NdS and EY carried out experiments. ZS, KL, and DY generated and provided the hESC mutant lines prior to publication, and provided significant consult. WKC provided patient-derived fibroblasts used for iPSC generation. All authors edited and agreed on the final manuscript version. This work was supported by grants from the National Institutes of Health to TE (R35HL135778), DH (R01DK096239), and WKC (P01HD068250), an NIH postdoctoral fellowship to JAB (F32HL152575), the MSK Cancer Center Support Grant/Core Grant (P30CA008748), the Tri-Institutional Medical Scientist Training Program to KMB (T32GM007739) and additional grants (TE and DH) from the Starr Tri-I Stem Cell Initiative (2016–004) and the New York State Department of Health (NYSTEM, C029567). Thank you

to Weill Cornell Medicine Genomics Resources Core Facility for assistance with RNA-seq and day 3 DEseq, and Piali Mukherjee and the Weill Cornell Medicine Epigenomics and Genomics Core Facilities for assistance with CUT&RUN and SEACR-based analysis.

## Additional information

### Competing interests

Kihyun Lee: Reviewing editor, eLife. Todd Evans: co-founder of OncoBeat, LLC. The other authors declare that no competing interests exist.

### Funding

| Funder | Grant reference number | Author |
|---|---|---|
| National Institutes of Health | R35135778 | Todd Evans |
| National Institutes of Health | R01DK096239 | Danwei Huangfu |
| National Institutes of Health | P01HD068250 | Wendy K Chung |
| National Institutes of Health | F32HL152575 | Joseph A Bisson |
| National Institutes of Health | P30CA008748 | Danwei Huangfu |
| Starr Foundation | 2016-004 | Danwei Huangfu Todd Evans |
| National Institutes of Health | T32GM007739 | Kelly M Banks |
| New York State Department of Health - Wadsworth Center | NYSTEM C029567 | Danwei Huangfu Todd Evans |

The funders had no role in study design, data collection and interpretation, or the decision to submit the work for publication.

### Author contributions

Joseph A Bisson, Conceptualization, Data curation, Formal analysis, Investigation, Writing - original draft, Writing – review and editing; Miriam Gordillo, Data curation, Investigation, Methodology; Ritu Kumar, Formal analysis, Investigation, Methodology; Neranjan de Silva, Ellen Yang, Investigation; Kelly M Banks, Data curation, Methodology; Zhong-Dong Shi, Kihyun Lee, Dapeng Yang, Wendy K Chung, Resources; Danwei Huangfu, Conceptualization, Resources, Project administration; Todd Evans, Conceptualization, Formal analysis, Supervision, Funding acquisition, Methodology, Project administration, Writing – review and editing

### Author ORCIDs

Joseph A Bisson ⓘ https://orcid.org/0009-0006-3702-7882
Todd Evans ⓘ https://orcid.org/0000-0002-7148-9849

### Ethics

This study uses human pluripotent stem cell lines including embryonic stem cell lines that are listed and approved on the NIH human embryonic stem cell registry. The study was reviewed and approved by the Tri-Institutional Stem Cell Initiative ESCRO.

Reviewer #1 (Public review): https://doi.org/10.7554/eLife.100797.3.sa1
Reviewer #2 (Public review): https://doi.org/10.7554/eLife.100797.3.sa2
Reviewer #3 (Public review): https://doi.org/10.7554/eLife.100797.3.sa3

Author response https://doi.org/10.7554/eLife.100797.3.sa4

## Additional files

### Supplementary files
Supplementary file 1. GATA6-RIME-enriched proteins at day 2 or day 4 of cardiac differentiation.

Supplementary file 2. Cell lines, primer sequences, and antibodies used in this study.

MDAR checklist

### Data availability
All RNA-seq and CUT&RUN data is available at the GEO repository and are publicly available as of the date of publication. The series accession numbers are GSE275685 and GSE275686.

The following datasets were generated:

| Author(s) | Year | Dataset title | Dataset URL | Database and Identifier |
|---|---|---|---|---|
| Bisson JA | 2024 | GATA6 regulates WNT and BMP programs to pattern precardiac mesoderm during the earliest stages of human cardiogenesis (CUT&RUN) | https://www.ncbi.nlm.nih.gov/geo/query/acc.cgi?acc=GSE275685 | NCBI Gene Expression Omnibus, GSE275685 |
| Bisson JA | 2024 | GATA6 regulates WNT and BMP programs to pattern precardiac mesoderm during the earliest stages of human cardiogenesis (RNA-Seq) | https://www.ncbi.nlm.nih.gov/geo/query/acc.cgi?acc=GSE275686 | NCBI Gene Expression Omnibus, GSE275686 |

The following previously published datasets were used:

| Author(s) | Year | Dataset title | Dataset URL | Database and Identifier |
|---|---|---|---|---|
| Loh KM, Chen A, Koh PW, Deng TZ, Sinha R, Tsai JM, Barkal AA, Shen KY, Jain R, Morganti RM | 2016 | Mapping the Pairwise Choices Leading from Pluripotency to Human Bone, Heart, and Other Mesoderm Cell Types | https://www.ncbi.nlm.nih.gov/geo/query/acc.cgi?acc=GSE85066 | NCBI Gene Expression Omnibus, GSE85066 |
| Teo AK, Arnold SJ, Trotter MW, Brown S, Ang LT, Chng Z, Robertson EJ, Dunn NR, Vallier L | 2011 | Pluripotency factors regulate definitive endoderm specification through eomesodermin | https://www.ncbi.nlm.nih.gov/geo/query/acc.cgi?acc=GSE77360 | NCBI Gene Expression Omnibus, GSE77360 |
| Tosic J, Kim GJ, Pavlovic M, Schroder CM, Mersiowsky SL, Barg M, Hofherr A, Probst S, Kottgen M, Hein L, Arnold SJ | 2019 | Eomes and Brachyury control pluripotency exit and germ-layer segregation by changing the chromatin state | https://www.ncbi.nlm.nih.gov/geo/query/acc.cgi?acc=GSE128466 | NCBI Gene Expression Omnibus, GSE128466 |
| Heslop JA, Pournasr B, Duncan SA | 2022 | Chromatin remodeling is restricted by transient GATA6 binding during iPSC differentiation to definitive endoderm | https://www.ncbi.nlm.nih.gov/geo/query/acc.cgi?acc=GSE156021 | NCBI Gene Expression Omnibus, GSE156021 |

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

# Appendix 1

## Appendix 1—key resources table

| Reagent type (species) or resource | Designation | Source or reference | Identifiers | Additional information |
|---|---|---|---|---|
| Gene (*Homo sapiens*) | *GATA6* | GenBank | GATA binding protein 6 | |
| Gene (*H. sapiens*) | *LRG5* | GenBank | Leucine-rich repeat containing G protein-coupled receptor 5 | |
| Strain, strain background (*Escherichia coli*) | One Shot TOP10 | Thermo Fisher | C404006 | Chemically competent cells |
| Cell line (*H. sapiens*) | H1 (NIH:0043) | WiCell | WA01 | Human embryonic stem cells |
| Cell line (*H. sapiens*) | Dermal fibroblast (normal, adult) | PCGC study patient | This study | Used to generate iPSC line |
| Cell line (*H. sapiens*) | iPSC line | Mutant GATA6 allelle c1071 delG | This study | See Data availability |
| Cell line (*H. sapiens*) | iPSC line | Allele reverted to wildtype | This study | See Data availability |
| Cell line (*H. sapiens*) | 293T | ATCC | CRL-11268 | |
| Biological sample (*Mus musculus*) | CF1 mouse embryonic fibroblasts | Thermo Fisher | A34180 | Primary cells |
| Transfected construct | pSpCas9n(BB)-2A-Puro (PX462) V2.0 | Addgene | 62987 | CRISPR vector for Nickase-Cas9 |
| Transfected construct (human) | CS-TRE-PRE-Ubc-tTA-I2G | Addgene | Derived from 198058 | Lentiviral vector for inducible expression of LGR5 |
| Antibody | Anti-human GATA6 (rabbit monoclonal) | Cell Signaling Technology | 5851 | WB (1:1000) RIME: 15 ug |
| Antibody | Anti-IgG control (rabbit monoclonal) | Cell Signaling Technology | 2729 | WB (1:1000) RIME: 15 ug |
| Antibody | See *Supplementary file 2* for a full list of antibodies | | | |
| Sequence-based reagent | gRNAs | Integrated DNA Technologies | See *Supplementary file 2* for a full list of guide RNAs | |
| Sequence-based reagent | qPCR primers | Integrated DNA Technologies | See *Supplementary file 2* for a full list of qPCR primers | |
| Sequence-based reagent | Donor sequence to correct mutant allele | Integrated DNA Technologies | GGCGCCACTGACGCCTG CCTGGCCCGCCGGACCC TTCGAGACCCCGGTGCT GCACAGCCTACAGAGCC GCGCCGGAGCCCCGC TCCCGGTG | See *Supplementary file 2* |
| Peptide, recombinant protein | bFGF | R&D Systems | 233-FB-025 | |
| Peptide, recombinant protein | Activin A | R&D Systems | 338-AC-050 | |
| Peptide, recombinant protein | BMP4 | R&D Systems | 314BP-050 | |
| Peptide, recombinant protein | VEGF | R&D Systems | 293-VE-050 | |
| Commercial assay or kit | CytoTune-iPS 2.0 Sendai Reprogramming Kit | Invitrogen | A16517 | For generating iPSC lines |
| Commercial assay or kit | T7 endonuclease I assay | New England Biolabs | M0302S | For indel detection |
| Commercial assay or kit | QuickExtract DNA Extraction Solution | Biosearch Technologies | QE09050 | DNA purification |
| Commercial assay or kit | SuperScript VILO cDNA Synthesis Kit | Invitrogen | 11754250 | cDNA generation |
| Commercial assay or kit | LightCycler 480 SYBR Green Master mix | Roche | 04887-352-001 | qPCR assays |

*Appendix 1 Continued on next page*

*Appendix 1 Continued*

| Reagent type (species) or resource | Designation | Source or reference | Identifiers | Additional information |
|---|---|---|---|---|
| Commercial assay or kit | Pierce Crosslink Magnetic IP/Co-IP Kit | Thermo Fisher | 88805 | Co-IP assays |
| Commercial assay or kit | TruSeq RNA Library Prep Kit v2 | Illumina | RS-122-2001 | Sequencing libraries |
| Commercial assay or kit | TruSeq Stranded mRNA Library Prep Kit | Illumina | 20020594 | Sequencing libraries |
| Commercial assay or kit | NEBNext Ultra II DNA Library Prep Kit | New England Biolabs | E7645 | CUT&RUN libraries |
| Commercial assay or kit | NEBNext Multiplex Oligos for Illumina | New England Biolabs | E6440 | CUT&RUN libraries |
| Commercial assay or kit | CUTANA ChIC/CUT&RUN Kit | Epicypher | 14-1048 | CUT&RUN |
| Commercial assay or kit | LookOut Mycoplasma PCR Detection Kit | Stem Cell Technologies | 1000276 | |
| Commercial assay or kit | Pierce BCA Protein Assay Kit | Thermo Fisher | 23225 | |
| Commercial assay or kit | Pierce ECL Western Blotting Substrate | Thermo Fisher | 32106 | |
| Chemical compound, drug | Y-27632 | Selleckchem | S1049 | ROCK inhibitor |
| Chemical compound, drug | Puromycin | Sigma-Aldrich | P8833 | Selection |
| Chemical compound, drug | XAV-939 | Sigma-Aldrich | X3004 | Tankyrase inhibitor |
| Chemical compound, drug | CHIR99021 | Stem Cell Technologies | 72054 | GSK-3 inhibitor |
| Chemical compound, drug | IWP2 | Tocris | 3533 | Porcupine inhibitor |
| Software, algorithm | ImageStudio | LI-COR | V5.2 | |
| Software, algorithm | Photoshop | Adobe | V25.9.1 | |
| Software, algorithm | bcl2fastq | Illumina | V2.19 | |
| Software, algorithm | STAR | Github | V2.5.2 | |
| Software, algorithm | Cufflinks | Github | V2.1.1 | |
| Software, algorithm | HTSeq-count | Github | V0.11.2 | |
| Software, algorithm | GSEA | Broad Institute | V4.2.3 | |
| Software, algorithm | Gene Ontology | Database for Annotation, Visualization, and Integrated Discovery | David.ncifcrf.gov | |
| Software, algorithm | SRplot | bioinformatics.com.cn/srplot | | |
| Software, algorithm | Prism | GraphPad | v.10.2.3 | |
| Other | Matrigel | Corning | BD354277 | Reagent |
| Other | StemFlex Medium | Thermo Fisher | A3349401 | Medium |
| Other | mTESR Plus Medium | Stem Cell Technologies | 1000276 | Medium |
| Other | mTESR1 medium | Stem Cell Technologies | 85850 | Medium |
| Other | Accutase Cell Detachment Solution | Selleckchem | S1049 | Reagent |
| Other | DMEM/F12 medium | VWR | 16777-255 | Medium |
| Other | KnockOut serum replacement medium | Thermo Fisher | 10828010 | Medium |
| Other | RPMI 1640 medium | Thermo Fisher | 2240089 | Medium |
| Other | B27 | Thermo Fisher | 17504044 | Medium additive |

*Appendix 1 Continued*

| Reagent type (species) or resource | Designation | Source or reference | Identifiers | Additional information |
|---|---|---|---|---|
| Other | B27-insulin | Thermo Fisher | A1895601 | Medium additive |
| Other | Sodium L-lactate | Sigma-Aldrich | L7022 | Medium additive |
| Other | Human albumin | Sigma-Aldrich | A9731 | Medium additive |
| Other | 1-Thioglycerol | Sigma-Aldrich | M6145 | Medium additive |
| Other | GlutaMAX | Thermo Fisher | 35050061 | Medium additive |
| Other | Transferrin | Roche | 10652202001 | Medium additive |
| Other | Trypsin/EDTA | VWR | 45000-664 | Reagent |
| Other | TrypLE Express Enzyme | Thermo Fisher | 12605010 | Reagent |
| Other | Saponin | Sigma-Aldrich | S7900-25G | Reagent |
| Other | RIPA Lysis Buffer | Thermo Fisher | 89900 | Reagent |
| Other | Protease/Phosphatase Inhibitor Cocktail | Cell Signaling Technology | 5872S | Reagent |
| Other | Lenti-X concentrator | Takara | 631231 | Assay kit |
| Other | Formaldehyde | Electron Microscopy Services | 15710 | Chemical |

