## [Editor Report · eLife Assessment]

This **important** study investigates the function of a critical regulator of human early cardiac development. The **convincing** examination of GATA6 function is thorough and well-executed. The study will be of interest to scientists working on how the human heart acquires its identity.

---

## [Referee Report · Reviewer #1 (Public review)]

Summary:

This is a comprehensive study that clearly and deeply investigates the function of GATA6 in human early cardiac development.

Strengths:

This study combines hESC engineering, differentiation, detailed gene expression, genome occupancy, and pathway modulation to elucidate the role of GATA6 in early cardiac differentiation. The work is carefully executed and the results support the conclusions. The use of publicly available data is well integrated throughout the manuscript. The RIME experiments are excellent.

Weaknesses:

Much has been known about GATA6 in mesendoderm development, and this is acknowledged by the authors.

Comments on revised version:

The authors have addressed my comments appropriately.

---

## [Referee Report · Reviewer #2 (Public review)]

Summary:

This manuscript by Bisson et al describes the role GATA6 to regulate cardiac progenitor cell (CPC) specification and cardiomyocyte (CM) generation using human embryonic stem cells (hESCs). The authors found that GATA6 loss-of-function hESC exhibit early defects in mesendoderm and lateral mesoderm patterning stages. Using RNA-seq and CUT&RUN assays the genes of the Wnt and BMP programs were found to be affected by the loss of GATA6 expression. Modulating Wnt and BMP during early cardiac differentiation can partially rescue CPC and CM defects in GATA6 hetero- and homozygous mutant hESCs.

Strengths:

The studies performed were rigorous and the rationale for the experimental designed were logical. The results obtained were clear and supports the conclusions that the authors made regarding the role of GATA6 on Wnt and BMP pathway gene expression.

Weaknesses:

Given the wealth of studies that have been performed in this research area previously, the amount of new information provided in this study is relatively modest. Nevertheless, the results and quite clear and should make a strong contribution to the field.

Comments on revised version:

The authors have addressed the prior request to assess genes expression representing each stage of development/differentiation from mesoderm to cardiac progenitor to cardiomyocytes and confirmed that the differentiation defect lies at the cardiac progenitor and cardiomyocyte stages and not in mesodermal differentiation. This work has significantly improved the robustness of the study.

---

## [Referee Report · Reviewer #3 (Public review)]

In this study, Bison et al. analyzed the role of the GATA6 transcription factor in patterning the early mesoderm and generating cardiomyocytes, using human embryonic stem cell differentiation assays and patient-derived hiPSCs with heart defects associated with mutations in the GATA6 gene. They identified a novel role for GATA6 in regulating genes involved in the WNT and BMP pathways. Modulation of the WNT and BMP pathways partially rescue early cardiac mesoderm defects in GATA6 mutant hESCs. These results provide significant insights into how GATA6 loss-of-function and heterozygous mutations contribute to heart defects.

Comments on revised version:

The authors have addressed all the concerns, using new data and modifications to the text to further strengthen the manuscript.

---

## [Author Response]

The following is the authors’ response to the original reviews.

**Public Reviews:**

**Reviewer #1 (Public review):**
Summary:This is a comprehensive study that clearly and deeply investigates the function of GATA6 in human early cardiac development.Strengths:This study combines hESC engineering, differentiation, detailed gene expression, genome occupancy, and pathway modulation to elucidate the role of GATA6 in early cardiac differentiation. The work is carefully executed and the results support the conclusions. The use of publicly available data is well integrated throughout the manuscript. The RIME experiments are excellent.Weaknesses:Much has been known about GATA6 in mesendoderm development, and this is acknowledged by the authors.

We appreciate the comments and have tried to highlight both the early role of GATA6 in cardiac progenitor biology as well as the haploinsufficiency for relevance to human congenital heart disease, which we believe adds value to other recent published work, among others Sharma et al. eLife 2020.

**Reviewer #2 (Public review):**
Summary:This manuscript by Bisson et al describes the role of GATA6 to regulate cardiac progenitor cell (CPC) specification and cardiomyocyte (CM) generation using human embryonic stem cells (hESCs). The authors found that GATA6 loss-of-function hESC exhibits early defects in mesendoderm and lateral mesoderm patterning stages. Using RNA-seq and CUT&RUN assays the genes of the Wnt and BMP programs were found to be affected by the loss of GATA6 expression. Modulating Wnt and BMP during early cardiac differentiation can partially rescue CPC and CM defects in GATA6 hetero- and homozygous mutant hESCs.Strengths:The studies performed were rigorous and the rationale for the experimental design was logical. The results obtained were clear and supported the conclusions that the authors made regarding the role of GATA6 on Wnt and BMP pathway gene expression.Weaknesses:Given the wealth of studies that have been performed in this research area previously, the amount of new information provided in this study is relatively modest. Nevertheless, the results and quite clear and should make a strong contribution to the field.

Likewise for reviewer 2, we appreciate the comments and have tried to highlight both the early role of GATA6 in cardiac progenitor biology as well as the haploinsufficiency for relevance to human congenital heart disease.

**Reviewer #3 (Public review):**
In this study, Bison et al. analyzed the role of the GATA6 transcription factor in patterning the early mesoderm and generating cardiomyocytes, using human embryonic stem cell differentiation assays and patient-derived hiPSCs with heart defects associated with mutations in the GATA6 gene. They identified a novel role for GATA6 in regulating genes involved in the WNT and BMP pathways -findings not previously noted in earlier analyses of GATA6 mutant hiPSCs during early cardiac mesoderm specification (Sharma et al., 2020). Modulation of the WNT and BMP pathways may partially rescue early cardiac mesoderm defects in GATA6 mutant hESCs. These results provide significant insights into how GATA6 loss-of-function and heterozygous mutations contribute to heart defects.I have the following comments:(1) Throughout the manuscript, Bison et al. alternate between different protocols to generate cardiomyocytes, which creates some confusion (e.g., Figure 1 vs. Supplemental Figure 2A). The authors should provide a clear justification for using alternative protocols.

We agree and clarified this issue in the revision (p. 6). The reviewer is correct that there are two widely used protocols for directed differentiation of PSCs to cardiac fate. One is a cytokine-based protocol (Fig. 1A) and the other uses small molecules to manipulate the WNT pathway (CHIR protocol, Supplemental Fig. 2B). In our study, we used the CHIR protocol only for experiments in Supplemental Figure 2B-E. Since our data implicated BMP and WNT as mediators of the GATA6-dependent program, we did this mainly to confirm that the phenotype we observed with the cytokine-based protocol was not biased by the differentiation protocol. However, we found the CHIR protocol to be overall relatively inefficient for cardiac differentiation using the parental H1 hESCs and the various isogenic lines. The in vitro cardiac differentiation protocols for hPSCs are known to be variable depending on lines and sometimes require extensive optimization for various media components and concentrations, cell seeding densities, and batch variations for crucial reagents. The cytokine-based protocol we optimized worked most efficiently with our hPSC lines to generate cardiomyocytes, therefore we committed to using it for the bulk of experiments in this study.

(2) The authors should characterise the mesodermal identity and cardiomyocyte subtypes generated with the activin/BMP-induction protocol thoroughly and clarify whether defects in the expression of BMP and WNTrelated gene affect the formation of specific cardiomyocyte subtypes in a chamber-specific manner. This analysis is important, as Sharma et al. suggested a role for GATA6 in orchestrating outflow tract formation, and Bison et al. similarly identified decreased expression of NRP1, a gene involved in outflow tract septation, in their GATA6 mutant cells.

We agree it is important that the mesodermal identities are quite thoroughly characterized.

For example, Fig. 2 (K+P+, Brachyury, EOMES), Fig. 3G&H (lateral mesoderm, cardiac mesoderm RNAseq & GSEA comparing datasets from Koh et al.). The capacity of the cytokine-based protocol to generate both FHF and SHF derived sub-types has been rigorously evaluated by Keller and colleagues, which we now cite (Yang et al. 2022). Since the null cells do not generate CMs, chamber specific subtypes cannot be evaluated; whether the GATA6 heterozygous mutants are biased is an interesting question. Indeed, the top GO term identified by CUT&RUN analysis for GATA6 at day 2 of

differentiation is outflow tract morphogenesis, which is consistent with the interpretation by Sharma et al., but implicates this program at a much earlier developmental stage, long before cardiomyocyte differentiation. We think this is one of the most important findings of our study and appreciate the chance to highlight this in the revision (p. 9, 17). When we evaluated chamber-specificity for differentiated cardiomyocytes, we did not find significant differences, as indicated for the reviewer in the panel below (day 20 of differentiation). Since our study focuses on early stages of progenitor specification rather than cardiomyocyte differentiation, we agree that a more rigorous analysis would be of value, and indicated this as a limitation of our current study (p. 18).

**Author response image 1. sa4fig1:** 

(3) The authors developed an iPSC line derived from a congenital heart disease (CHD) patient with an atrial septal defect and observed that these cells generate cTnnT+ cells less efficiently. However, it remains unclear whether atrial cardiomyocytes (or those localised specifically at the septum) are being generated using the activin/BMP-induction protocol and the patient-derived iPSC line.

As indicated above, our study is focused on cardiac progenitor specification, and we found similar differences with the patient-derived iPSC-CMs compared to using hESC heterozygous targeted mutants. While we did not note any major differences in expression of cardiomyocyte markers, whether the mutants show any biases toward sub-types of cardiomyocytes is an interesting question to be pursued in subsequent work.

(4) The authors should also justify the necessity of using the patient-derived line to further analyse GATA6 function.

This is a good point, and as suggested we provided the justification (p. 5-6). This is the first patient-derived iPSC line published with a heterozygous GATA6 mutation along with an isogenic mutation-corrected control generated for cardiac directed differentiation. Patients with congenital heart disease (CHD) associated with GATA6 mutations are typically heterozygous (also true for many other CHD variants; presumably homozygous null embryos would not survive). It is important to query if phenotypes found using targeted mutations in hESCs (or iPSCs) model the human disease, since the patient cells (or the hESCs) likely have additional genetic variants that might interact with the GATA6 mutation. The fact that both types of heterozygous cells (patient-derived iPSCs and targeted hESCs) generate similar defects in CM differentiation provides evidence supporting the use of these human cellular models to study the genetic and cellular basis for congenital heart disease. This is particularly important, since other models, such as heterozygous mice, do not show such phenotypes.

(5) Figure 3 suggests an enrichment of paraxial mesoderm genes in the context of GATA6 loss-of-function, which is intriguing given the well-established role of GATA6 in specifying cardiac versus pharyngeal mesoderm lineages in model organisms. Could the authors expand their analysis beyond GO term enrichment to explore which alternative fates GATA6 mutant cells may acquire? Additionally, how does the potential enrichment of paraxial mesoderm, rather than pharyngeal mesoderm, relate to the initial mesodermal induction from their differentiation protocol? Could the authors also rule out the possibility of increased neuronal cell fates?

We need to interpret our in vitro differentiation data cautiously in relation to what has been shown in vivo, since we are unlikely to be reproducing all the complex signaling taking place in the embryo. Yet we do see modest increases in gene expression levels including signatures of paraxial mesoderm and ECM/mesenchymal at days 2 or 3 of differentiation in the GATA6 mutant cells. Therefore, we now include a heatmap showing enriched paraxial mesoderm gene expression in the mutant cells, new Fig. 3I (see page 10).

A caveat of this result is that the cells are being differentiated toward cardiac fate, so a bias for alternative fates might be suppressed. We modified the protocol to favor paraxial fate by adding CHIR at day 2 (rather than XAV) and performing qPCR assays at day 3. We found this successfully induced paraxial mesoderm gene expression, but equally comparing wildtype, heterozygous, or null cells, so do not feel it warrants highlighting further.

**Recommendations for the authors:**

**Reviewing Editor (Recommendations for the authors):**
Incorporation of marker analysis for various stages of iPSC to CM differentiation (mesoderm, cardiac progenitor, CM subtypes) would increase the significance and support for the findings presented. Further data on the link (direct or indirect) between GATA6 and Wnt/BMP signalling would also add to the significance of this study. A number of textual changes/clarifications are also suggested to improve the manuscript.

We appreciate the feedback and provide responses for issues raised for markers, direct or indirect interactions, and textual changes/clarifications in the following sections. As indicated above, we did not find obvious alterations in cardiac subtypes, but since our study is focused on early progenitor specification, this is an interesting question that we think should be more rigorously evaluated in subsequent work.

**Reviewer #1 (Recommendations for the authors)**:Minor details:(1) On p6 "Principal component analysis (PCA) showed that the cells derived from each genotype were well separated from each other (Supplemental Figure 2C)". All genotypes should be in one PCA plot to better evaluate the three genotypes.

We prepared the new plot as suggested, presented as new Supplemental Fig. 2C.

(2) p10: "Chia et al.22 and found a significantly decreased enrichment in GATA6-/- cells relative to WT at day 2" decreased enrichment of what? Direct target genes?

Thank you for catching this. Yes, the text was changed to indicate a “decreased enrichment in GATA6-/- cells relative to WT at day 2 for putative direct GATA6 target genes.”

**Reviewer #2 (Recommendations for the authors):**
Overall, this is an interesting study that addresses the early developmental roles of GATA6 on cardiac differentiation. While the identification of Wnt and BMP pathway genes to be involved in GATA6 regulation is not entirely unexpected, the authors do bring forth some useful knowledge that helps to further elucidate the mechanism of pre-cardiac mesoderm regulation. Some suggestions for improvement are included below -Major points:(1) Since the loss of Gata6 in this study is global (either as heterozygous or homozygous, it is likely that the very early requirement of Gata6 (e.g. mesodermal stage of differentiation)) is responsible for the cardiac transcriptional phenotype observed and not due to specific role of Gata6 in the cardiac lineage which would need to be addressed using conditional knock out of Gata6 in hPSC model. The authors should be more explicit when discussing the results as disruption of mesodermal differentiation leading to loss of downstream cardiac lineage cells. For example, I would change the title "GATA6 loss-of-function impairs CM differentiation" to "GATA6 loss-of-function impairs mesodermal (or mesodermal lineage) differentiation" and show the changes in cardiac progenitor cells genes (Isl1, Tbx1, Hand1, and BAF50c/Smarcd3) in addition to cardiomyocyte genes but no change in mesodermal (e.g. Brachyury, T, Eomes, Mesp1/2, etc) genes.

We agree with the reviewer’s interpretation. The title for the section was changed as suggested. In Fig. 1, we show changes in cardiac progenitor cell genes (Isl1, Hand1, and BAF50c/Smarcd3) while not seeing changes in mesodermal genes in Fig. 2 (e.g. Brachyury, Eomes, Mesp1/2). We note that the defect may be specific to cardiac (or anterior lateral) mesoderm, as the ability to express paraxial mesoderm markers was not impaired.

(2) The use of NKX2.5, TBX5, TBX20, and GATA4 as markers for CPC is not ideal. These markers are also expressed in differentiated cardiomycytes. ISL1 or TBX1 for second heart field progenitors and HAND1 or BAF60c/Smarcd3 for first heart field progenitors would be ideal.

As suggested, we included additional day 6 qPCR panel (new Fig. 1E) to evaluate the heart field progenitor markers.

(3) Much of the findings described in this study have been known in the field including the requirement of Wnt and BMP to induce mesodermal and subsequently cardiomyocyte differentiation. The key new information here is that Gata6 knockout disrupts Wnt and BMP signaling. It would help to further validate experimentally some of the Wnt and BMP genes as either direct or indirect targets of Gata6 using reporter assays.

While reporter assays are feasible and do provide relevant outputs, we feel that the use of any one or even several response elements in a reporter assay adds relatively little value compared to comprehensive analysis of bona fide network components. To address the reviewers concern we have included profiling heat maps for WNT and BMP pathway components to more rigorously and specifically evaluate the disruption in the signaling networks caused by loss of GATA6. Proving direct targets of endogenous genes is challenging, but we mapped many binding peaks for GATA6 to putative enhancers of WNT/BMP pathway genes (based on histone marks). We provide a list of these genes (new Fig. 4F) and distinguish these from WNT/BMP pathway genes that were not bound by GATA6 yet are down-regulated in the GATA6 mutant cells and are likely to be indirect targets (p. 12).

Minor points:(1) Figures 1 and 2 - in the figure legend the labels w2, w4, m2, m5, m11, and m14 should be explained as the name of the clones of targeted hESC.

The legends were edited to provide this information.

(2) Supplemental Figure 3A - the resolution of the FACS plot is suboptimal.

We apologize and have corrected the plot resolution in the revised manuscript.

(3) Supplemental Table 1 - it's intriguing that amongst all the SWI/SNF factors, the one that is known to be cardiac-specific (SMARCD3) did not come up in the GATA6-RIME-enriched proteins. Is this a reflection of the early stage in which GATA6 plays a role in development (e.g. mesendoderm development but not precardiac mesoderm development when SMARCD3 is expressed)?

We agree and have noted this feature in the revised manuscript (p. 17). We note that SMARCD3 is expressed in the RNA-seq data as early as day 2. Although speculative, it may be that GATA6 primarily interacts with SWI/SNF complexes prior to the role for SMARCD3 in cardiac specification.

**Reviewer #3 (Recommendations for the authors):**
(1) Figures 3G and 3H, as well as others, have resolution issues. The gene names are unreadable, and higherresolution images should be provided.

We apologize for the resolution issues and these have been fixed in the revised version.

(2) In their early manipulation of the WNT and BMP pathways (Figure 6A), it is unclear whether the activin/BMP protocol shown in Figure 1A was used. If this is the case, the authors should compare their results to a wild-type + DOX EV condition for consistency.

We clarified in the revision (Fig. 6A) that all the experiments in Fig. 6 use the cytokine protocol. In the revised figure, we included the wild-type + DOX EV condition as suggested.

(3) In Figures 6C and 6D, the authors should include an analysis of a wild-type isogenic line under their new CHIR/LB condition for comparison.

As suggested, we included the WT isogenic line in the comparison. For Fig. 6C these are shown on a separate graph because the Y-axis values are very different. Note that the CHIR/LB treatments that improve mutant cell differentiation impact the WT cells in the opposite manner.